# Dynamic response of Antarctic Peninsula Ice Sheet to potential collapse of Larsen C and George VI ice shelves

Clemens Schannwell[1,2], Stephen Cornford[3], David Pollard[4], and Nicholas Edward Barrand[1]

[1]School of Geography, Earth and Environmental Sciences, University of Birmingham, Birmingham, UK
[2]British Antarctic Survey, Natural Environment Research Council, Cambridge, UK
[3]Centre for Polar Observation and Modelling, School of Geographical Sciences, University of Bristol, Bristol, UK
[4]Earth and Environmental Systems Institute, Pennsylvania State University, University Park, PA, USA

*Correspondence to:* Clemens Schannwell (Clemens.Schannwell@uni-tuebingen.de)

**Abstract.** Ice shelf break-up and disintegration events over the past five decades have led to speed-up, thinning, and retreat of upstream tributary glaciers and increases to rates of global sea-level rise. The southward progression of these episodes indicates a climatic cause, and in turn suggests that the larger Larsen C and George VI ice shelves may undergo similar collapse in the future. However, the extent to which removal of Larsen C and George VI ice shelves will affect upstream tributary glaciers and add to global sea levels is unknown. Here we apply numerical ice-sheet models of varying complexity to show that the centennial sea-level commitment of Larsen C embayment glaciers following immediate shelf collapse is low (<2.5 mm to 2100, <4.2 mm to 2300). Despite its large size, Larsen C does not provide strong buttressing forces to upstream basins and its collapse does not result in large additional discharge from its tributary glaciers in any of our model scenarios. In contrast, the response of inland glaciers to collapse of George VI Ice Shelf may add up to 8 mm to global sea levels by 2100 and 22 mm by 2300 due in part to the mechanism of marine ice sheet instability. Our results demonstrate the varying and relative importance to sea level of the large Antarctic Peninsula ice shelves considered to present a risk of collapse.

## 1 Introduction

The observational history of ice-shelf collapse in the Antarctic Peninsula has led to a proposed northerly limit of ice-shelf viability determined by the -9°C mean annual isotherm (Mercer, 1978; Morris and Vaughan, 2003). Recent, rapid warming has led to the southward migration of this limit (Vaughan et al., 2003), now threatening the stability of the large Larsen C and George VI ice shelves. The northernmost remaining ice shelf (Figure 1a), Larsen C, is considered to present the greatest risk of collapse (Jansen et al., 2015). While other mechanisms such as ice-shelf thinning, fracturing, and weakening of shear margins may contribute to Larsen C ice-shelf instability (Kulessa et al., 2014; Holland et al., 2015; Borstad et al., 2016), the risk of shelf collapse has increased slightly since summer 2017 when a large iceberg calved off Larsen C. This calving event leaves Larsen C in conditions similar to those present immediately prior to the collapse of Larsen B Ice Shelf in 2002 and may promote instability (Jansen et al., 2015).

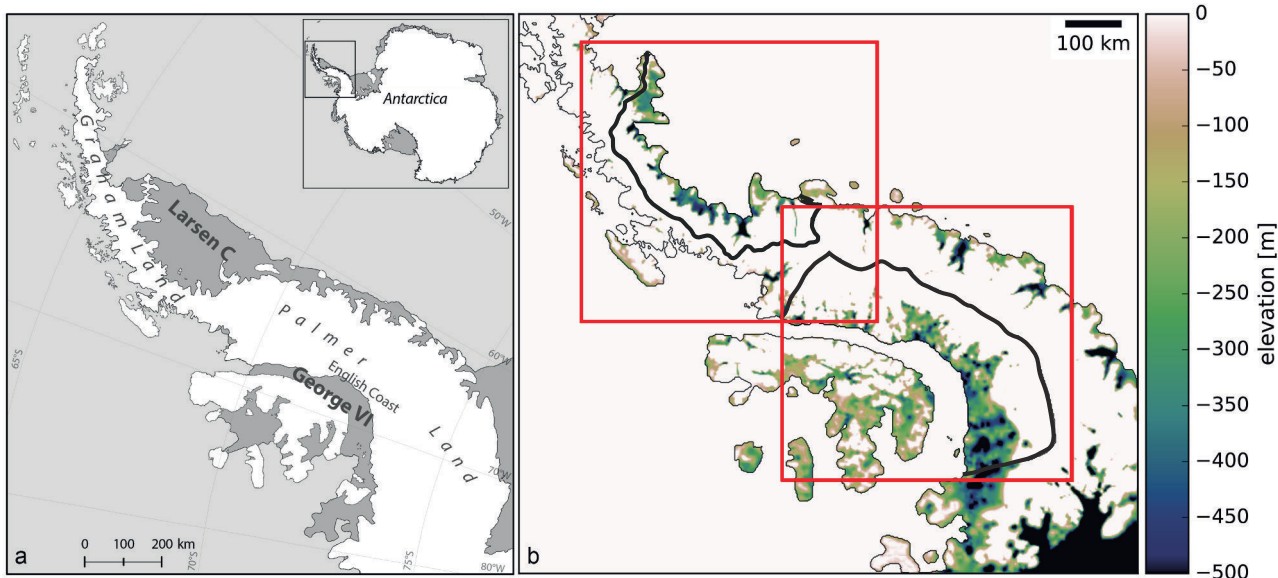

**Figure 1.** (a) Location map of the Antarctic Peninsula including locations of Larsen C and George VI ice shelves and localities mentioned in the text. (b) Bedrock elevations below sea level in meters for the Antarctic Peninsula from BEDMAP2 (Fretwell et al., 2013). The colourbar is truncated at 0 m. Red inset rectangles delineate location of zoom-in views in Figures 2,5,6, and 8. Black polygons denote ice-sheet model domains.

Despite the increased research focus on Larsen C Ice Shelf, most of the current mass loss and contribution to sea-level rise from the Antarctic Peninsula originates from large drainage basins feeding George VI ice shelf, along the English Coast, western Palmer Land, in the south-west of the peninsula (McMillan et al., 2014; Martín-Español et al., 2016). Here, outlet glaciers have thinned rapidly in the last two decades, contributing $\sim$0.1 mm a$^{-1}$ to global sea-level rise (Wouters et al., 2015; Hogg
et al., 2017). Many of these glaciers are grounded below sea-level with deeply-incised bedrock troughs and retrograde sloping bedrock topography (Figure 1b). These marine-based sectors, which contain a sea-level equivalent of 46.2 mm (25% of the total ice volume in the APIS, Figure 1b), are therefore potentially vulnerable to the marine ice sheet instability mechanism, a tendency of grounding-line retreat to accelerate in the absence of compensating forces (Schoof, 2007; Gudmundsson et al., 2012).

Here we use three ice-sheet models of varying complexity to compute the upstream glacier response and sea-level rise commitment following potential collapse of Larsen C and George VI ice shelves. Owing to differences in model setup and physics, this study does not provide a full model intercomparison, but rather presents a multi-model spread sea-level envelope assessment using a range of ice-flow approximations: (i) the linearised shallow-ice approximation (SIA) model BAS-APISM (Barrand et al., 2013); (ii) the hybrid sheet-shelf model PSU3D (Pollard and DeConto, 2012a), and; (iii) the vertically-integrated sheet-
shelf model BISICLES (Cornford et al., 2013). This multi-model approach provides a starting point for regional ice-sheet model forecasts and sea-level impact studies and allows examination of process differences in glacier responses across the drainage basins of Larsen C and George VI ice shelves.

## 2 Methods

The ice-sheet models BAS-APISM (Barrand et al., 2013), BISICLES (Cornford et al., 2013), and PSU3D (Pollard and De-Conto, 2012a) have been described in detail elsewhere. A summary of model description, parameterisation and experimental design relevant to this study are presented here, including important changes to model setups from previously published configurations. Model domains vary across the models with BAS-APISM and BISICLES including the entire Antarctica Peninsula and PSU3D simulating the Larsen C embayment and George VI embayment separately (red rectangles in Figure 1).

### 2.1 Ice-sheet model description

BAS-APISM (Barrand et al., 2013) simulates ice flow by solving the simplest permissible force balance approximation - the linearised shallow-ice approximation (SIA). Owing to the linearisation, the model is less sensitive to ice thickness errors than traditional SIA-based models. The linear nature of the model equations permits simple summation of sea-level rise contributions from individual drainage basins to provide an ice-sheet wide estimate. As the SIA is not valid for floating ice shelves (Hutter, 1983), only the grounded ice sheet is simulated and grounding-line retreat is parameterised through a statistical model. This model scales the expected retreat of the grounding line in response to ice-shelf collapse to the amount of buttressing at the ice front of each drainage basin (Schannwell et al., 2016). Ice-shelf buttressing was computed from output of an ice-sheet model inversion (Arthern et al., 2015). As BAS-APISM cannot simulate grounding-line advance, ice-shelf flow, or ice-shelf buttressing, this model is only employed in Experiment 1 (immediate ice-shelf collapse) where ice-shelf flow is not explicitly simulated (See Section 2.5) and immediate ice-shelf collapse is assumed.

PSU3D (Pollard and DeConto, 2012a) simulates ice flow by using a hybrid combination of the scaled SIA and shallow-shelf approximation (SSA) equations. The SSA is valid for ice shelves and ice streams characterised by low basal drag. This type of ice-sheet model (A-HySSA = asymptotic hybrid SIA-SSA model (Pattyn et al., 2013)) provides the required physics to simulate the ice sheet-ice shelf system, including explicit tracking of the position of the grounding line. To make the model less sensitive to grid resolution, an additional internal flux boundary condition is employed at the grounding line. The model set-up used here is similar to Pollard et al. (2015), but cliff failure and bedrock deformation are not included. PSU3D solves the time varying 3-D temperature equation, but surface air temperature forcing is held constant at year 2000 (Le Brocq et al., 2010) throughout the simulations.

BISICLES (Cornford et al., 2013) simulates ice flow by solving a vertically integrated stress balance (L1L2 = one-layer longitudinal stress model (Hindmarsh, 2004)) to determine the horizontal velocity. The ice rheology is given by Glen's flow law

$$\boldsymbol{S} = 2\phi\eta\dot{\boldsymbol{\epsilon}}. \tag{1}$$

Here $\boldsymbol{S}$ is the deviatoric stress tensor, $\eta$ is the effective viscosity , $\dot{\boldsymbol{\epsilon}}$ is the strain-rate tensor and $\phi$ is the stiffening factor that accounts for ice damage, anisotropy, and temperature uncertainties (Cornford et al., 2015). This type of stress balance is similar to the SSA, but includes vertical shearing in the effective viscosity calculation, resulting in softer ice at the grounding line in

comparison to traditional SSA models and resembles more the behaviour of full-Stokes models (Pattyn and Durand, 2013). The equations are solved on an adaptive 2-D grid, allowing for higher resolution in areas of interest such as grounding lines or shear margins, and coarser resolution away from these regions to save computation time. A subgrid interpolation scheme for basal drag near the grounding line was employed to improve the accuracy of the grounding-line position at each time step (Cornford

et al., 2016). In all BISICLES simulations ice temperature data are provided by a three-dimensional thermo-mechanical model (Pattyn, 2010) and is held fixed in time.

Basal traction in PSU3D and BISICLES is determined by a viscous law

$$\tau^b = \begin{cases} -C|\boldsymbol{u}|^{m-1}\boldsymbol{u} & \text{if } \frac{\rho_i}{\rho_w}h > -b, \\ 0 & \text{otherwise,} \end{cases} \tag{2}$$

where $m$=0.5 (quadratic law), $\tau^b$ is the basal traction, $\boldsymbol{u}$ is the horizontal velocity, $\rho_i$ and $\rho_w$ are ice and ocean densities, $b$ is the

bedrock elevation, $h$ is ice thickness, and $C$ is the basal friction parameter inferred by solving an inverse problem (See Section 2.3). Due to the linearisation of the evolution equations in BAS-APISM, there is no need to specify whether or not basal sliding is occurring. All rates are determined by the ice flux which is directly derived from the data (Barrand et al., 2013).

Basal sliding sensitivity simulations with BISICLES were also performed with $m$=1/3 (cubic law) and $m$=1 (linear law). In addition, a simulation was performed using a Coulomb-limited law (Tsai et al., 2015). This law combines the power law (Equa-

tion 2) with the Coulomb friction law by ensuring that basal traction cannot exceed the Coulomb friction that is proportional to the effective pressure $N_e$:

$$|\tau_b| = \min\left(aN_e, C|\boldsymbol{u}|^m\right), \tag{3}$$

where the first term in the parentheses is the Coulomb friction law with $a$=0.5, $m$=0.5 and the effective pressure $N_e$ is

$$N_e = \rho_i g(h - h_f), \tag{4}$$

where $g$ is the acceleration of gravity and $h_f$ is the flotation thickness. Equation 4 is only valid under the assumption of full connection between the basal hydrology and the ocean. Since the Coulomb law implies that basal drag approaches zero towards the grounding line, this type of basal sliding law ensures a smooth transition from grounded to floating ice, unlike the traditional power law (Equation 2) which implies that basal drag is highest near the grounding line (Tsai et al., 2015).

## 2.2  Calving

In simulations where the calving front is not fixed e.g. where ice-shelf flow and retreat is explicitly simulated, calving depends on the depths of surface ($d_s$) and basal crevasses ($d_b$), relative to total ice thickness. Crevasse depths are computed by (Benn et al., 2007; Nick et al., 2010)

$$d_s = \frac{2}{\rho_i g}\left(\frac{\dot{\epsilon}}{\bar{A}}\right)^{\frac{1}{n}} + \frac{\rho_w}{\rho_i}d_w, \tag{5}$$

$$d_b = \left( \frac{\rho_i}{\rho_0 - \rho_i} \right) \frac{2}{\rho_i g} \left( \frac{\dot{\epsilon}}{\bar{A}} \right)^{\frac{1}{n}}, \tag{6}$$

where $\bar{A}$ is the depth-averaged rheological coefficient, $n=3$ is the rheological exponent, $d_w$ is the water height in the surface crevasse and $\rho_0$ is the density of surface water. The parameter $\dot{\epsilon}$ is the longitudinal strain rate approximated in PSU3D through the isotropic ice divergence

$$\dot{\epsilon} = \left( \frac{\partial u}{\partial x} + \frac{\partial v}{\partial y} \right). \tag{7}$$

BISICLES implements essentially the same criterion, but computes crevasse depths from membrane stresses such that Equations 5-6 become

$$d_s = \frac{\mathrm{tr}(\tau)}{\rho_i g} h + \frac{\rho_w}{\rho_i} d_w, \tag{8}$$

$$d_b = \frac{\rho_i}{\rho_0 - \rho_i} \left( \frac{\mathrm{tr}(\tau)}{\rho_i g} h - h_{ab} \right), \tag{9}$$

where $\tau$ is the deviatoric stress tensor, $\mathrm{tr}()$ is the trace operator, and $h_{ab}$ is the thickness above flotation (Sun et al., 2017). In PSU3D ice is calved off when the combined ice thickness of the surface and bottom crevasses reach at least 75% of the column ice thickness (Pollard et al., 2015), whereas in BISICLES icebergs calve when the sum of the surface and bottom crevasses reaches the distance from ice surface to the waterline.

Water height in surface crevasses ($d_w$ in Equation 5) is computed from biased-corrected CMIP5 model projections to 2300 from the model selection presented in Schannwell et al. (2015). The bias-correction and melt computation approach follows Trusel et al. (2015). In brief, December-January-February (DJF) near-surface temperatures from the CMIP5 historical simulations were compared to high resolution (5.5 km) RACMO2.3 simulations (van Wessem et al., 2016) such that

$$\overline{Bias_{GCM}} = \overline{T_{2m_{GCM}}} - \overline{T_{2m_{RACMO2.3}}}, \tag{10}$$

where $\overline{T_{2m_{GCM}}}$ and $\overline{T_{2m_{RACMO2.3}}}$ are the mean DJF near-surface temperatures over the baseline period 1980-2005 from each GCM and RACMO2.3, respectively. The bias calculation (Equation 10) is restricted to the ice-shelf areas in our two model domains (Figure 1). The best performing GCM (lowest bias) for the RCP4.5 (Figure A1, MIROC-ESM) and RCP8.5 (Figure A2, CSIRO) scenarios were then selected as future forcing.

To convert from near-surface temperature to melt, the empirical formula derived by Trusel et al. (2015) was used. This formula scales surface melt ($R$ in equation 11) exponentially with mean DJF near-surface temperatures and approximates the surface melt available to fill surface crevasses. To compute water height in surface crevasses, $d_w$ is set to (Pollard et al., 2015):

$$d_w = 100R^2. \tag{11}$$

## 2.3 Model Initialisation

BAS-APISM employs a combined altimetric and velocity initialisation scheme, permitting a steady-state starting condition after initialisation under the assumption that the current ice sheet configuration is close to steady state (Barrand et al., 2013). This is accomplished through the computation of balance fluxes. The motivation for this type of initialisation technique is that the absence of accurate ice thickness datasets leads to the omission of the mechanical model in the cost function employed for the initialisation (Barrand et al., 2013).

BISICLES is initialised by solving an optimisation problem to infer the basal traction coefficient $C$ and the stiffening factor $\phi$ (also enhancement factor, Equation 1), by matching modelled velocities with observed velocities (Rignot et al., 2011). This type of initialisation is well known and widely employed in ice sheet modelling (MacAyeal, 1992; Cornford et al., 2015). A nonlinear conjugate gradient method was employed to seek a minimum of the objective function

$$J = J_m + J_p \tag{12}$$

where $J_m$ is the misfit between observed and modelled velocities and $J_p$ is a Tikhonov penalty function described by

$$J_p = \lambda_C J_C^{reg} + \lambda_\phi J_\phi^{reg} \tag{13}$$

where $\lambda_C$ and $\lambda_\phi$ are the Tikhonov parameters and $J_C^{reg}$ and $J_\phi^{reg}$ represent the spatial gradients of $C$ and $\phi$ integrated over the domain (Cornford et al., 2015). An L-curve analysis was performed to calibrate the Tikhonov parameters and avoid overfitting or overregularisation (Fürst et al., 2015). The selected values are $\lambda_C = 10^{-1}$ and $\lambda_\phi = 10^9$ (Figure A3).

In solving this inverse problem, maps of surface elevation and bedrock topography were taken from the BEDMAP2 (Fretwell et al., 2013) dataset, and a steady state 3-D temperature field was used from a higher order model (Pattyn, 2010). It is only necessary to find solutions with a single sliding law, as the coefficients can be computed from one another to give the same basal traction $\tau_b$, e.g the coefficients for the cases $m = m_1$ and $m = m_2$ must satisfy $C_2|u|^{m_2} = C_1|u|^{m_1}$. We chose $m = 1$ for the inversion simulation.

PSU3D utilises a different algorithm to infer the basal traction coefficient. Instead of matching velocities, the algorithm implemented in PSU3D seeks to minimise the misfit between local surface elevation observations and modelled local surface elevations (Pollard and DeConto, 2012b). To achieve this, the ice-sheet model is run forward in time, and basal traction coefficients are periodically compared and adjusted according to the local surface elevation error. This iterative process is continued until modelled surface elevation converges to the best fit with observed surface elevation (Pollard and DeConto, 2012b). Note that this simpler algorithm does not infer a stiffening factor $\phi$ for ice shelves. Input maps needed for the inversion algorithm are from ALBMAP (Le Brocq et al., 2010): e.g., ice thickness and bedrock topography. For all PSU3D simulations, basal traction fields are interpolated onto the respective model grid from a 5 km Antarctica inversion simulation. The coarser resolution leads to some interpolation artefacts in the basal traction coefficient fields (Figure 2).

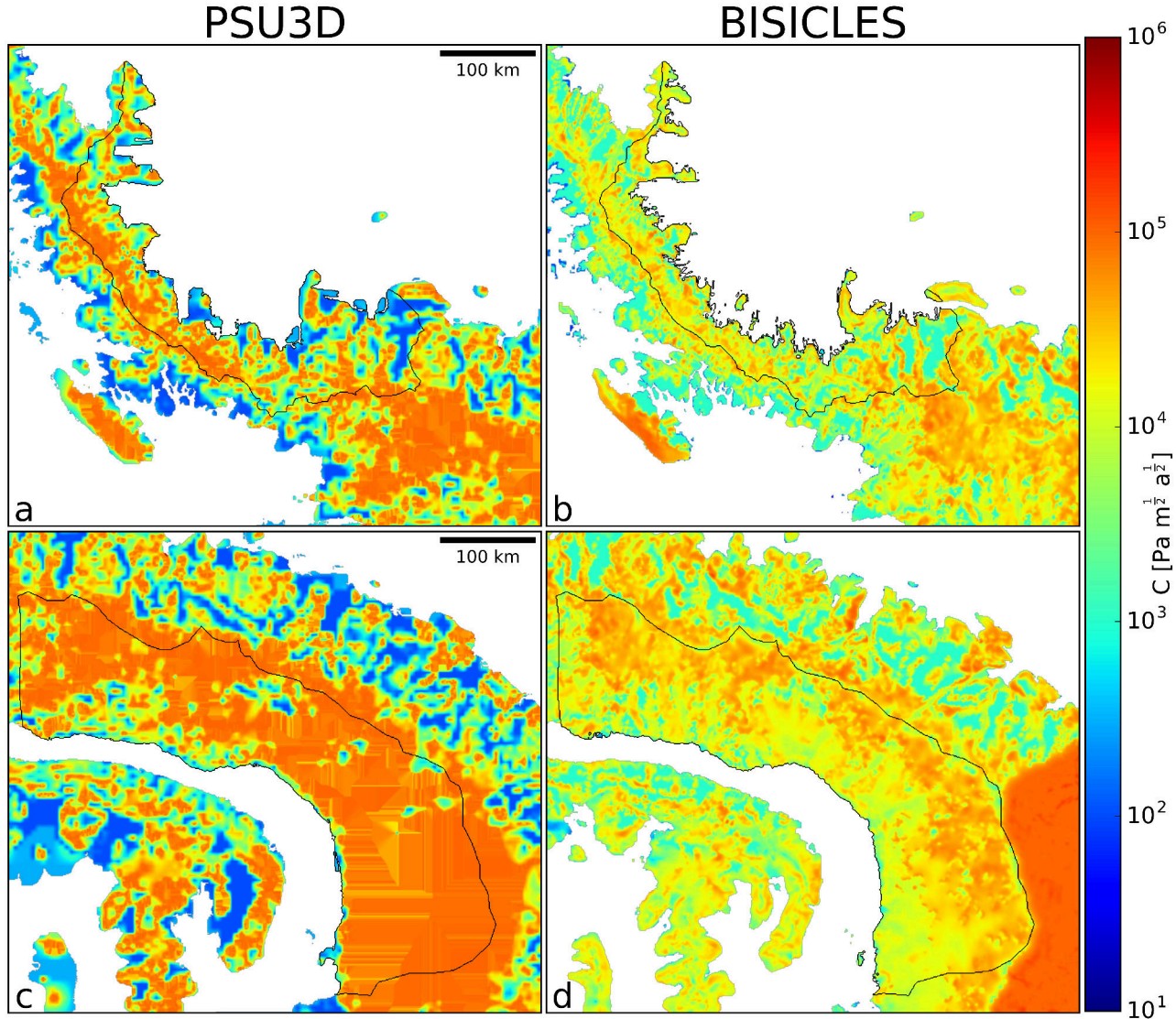

**Figure 2.** Inferred basal traction fields $C$ for the Larsen C (a,b) and George VI model domains (c,d). Black lines denote modelled drainage basins.

## 2.4 Spin-Up

Following initialisation, the sheet-shelf models should aim to be as close to the steady-state initial conditions provided by observations, as long as the ice sheet itself is in steady state, such that $\frac{\partial h}{\partial t} = 0$. However, owing to data inconsistencies and in part a violation of this steady-state assumption this condition is not fulfilled, requiring a spin-up or relaxation simulation to reach a steady state for each model. To tease out the sea-level rise contributions from ice-shelf removal and facilitate comparison

across all three ice-sheet models, the employed spin-up approach aims to keep the ice sheet geometry as close as possible to the initial geometry. This is necessary because BAS-APISM provides a stable starting condition after initialisation. To ensure a minimal change in ice-sheet geometry, we compute a synthetic mass balance (MB) which is simply (Price et al., 2017)

$$MB = FC, \tag{14}$$

5    where $FC$ is the negative of the modelled thickness field change when the model is run forward a single time step. This synthetic mass balance is applied in all spin-up and perturbation simulations. All simulations are then run forward in time for 50 years with only this forcing applied. To reach steady state, the volume above flotation change with time should be near zero ($\frac{\partial V}{\partial t} \sim 0$) at the end of the spin-up. All of our simulations fulfil this criterion (Figure 3), even though PSU3D simulations are not as close to steady state as BISICLES simulations at the end of the spin-up period.

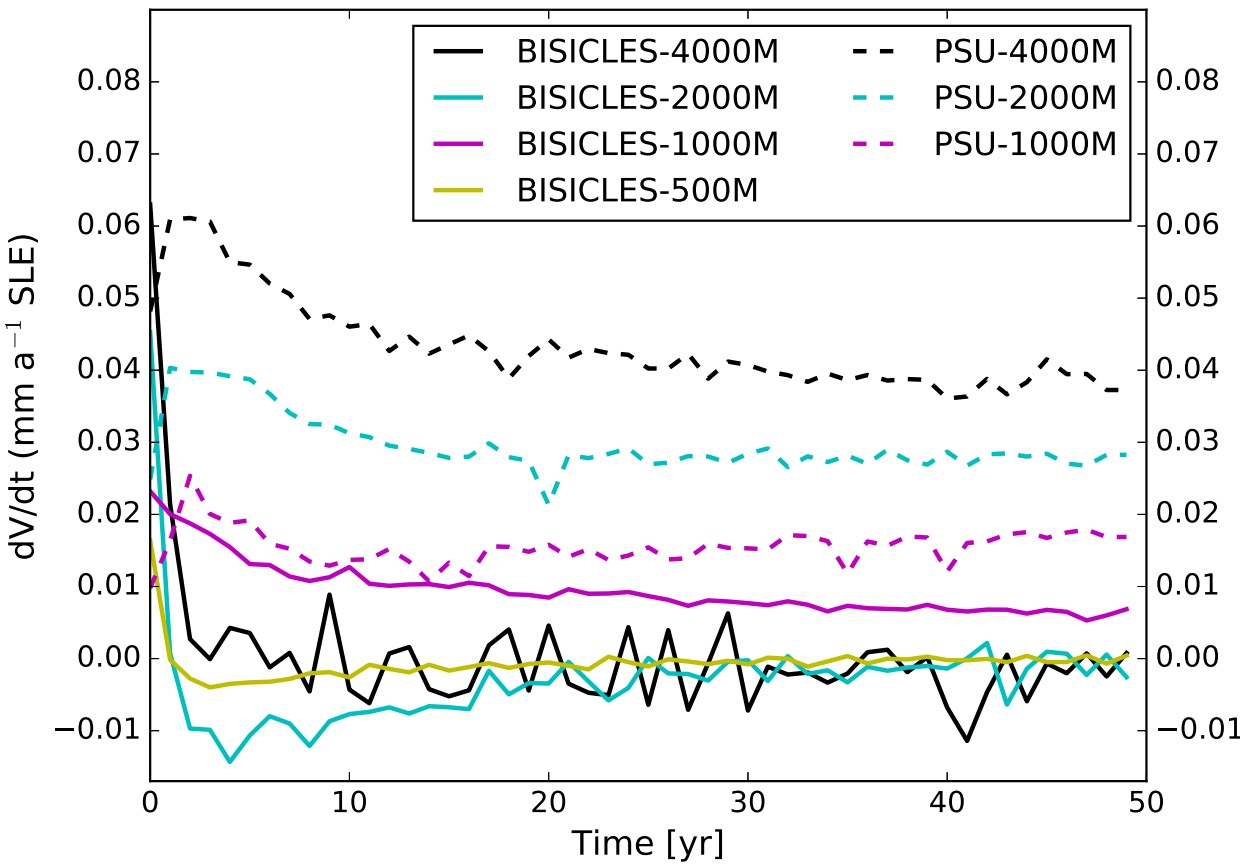

**Figure 3.** Volume change ($\frac{\partial V}{\partial t}$) spin-up plot for BISICLES (solid lines) and PSU3D (dashed lines) at different horizontal resolutions.

## 2.5 Experimental Design

Two sets of experiments were undertaken using the ice-sheet models (Table 1). In Experiment 1, immediate ice-shelf collapse was imposed on all three ice-sheet models and combined with a fixed calving front position. This provides an envelope of sea-level rise projections for the peninsula region and evaluates the importance of each shelf to the tributary glaciers upstream.
Simulations with the sheet-shelf models (PSU3D and BISICLES) were carried out at different horizontal resolutions to investigate the grid dependence on the sea-level rise projections and to select the best compromise between computational demand and appropriate grid resolution (Table 1). In the second simulation (Experiment 2), the two sheet-shelf models (PSU3D and BISICLES) were run at 1 km resolution to simulate ice-shelf retreat and collapse and subsequent tidewater glacier retreat using a physically-based calving relation (Benn et al., 2007; Nick et al., 2010). This relation initiates iceberg calving when the com-
bined depth of surface and bottom crevasses reach a threshold percentage of ice thickness (See Section 2.2). Crevasse depth primarily depends on the stress field of the ice shelf, with extensional stresses providing favourable conditions for crevasse opening, though meltwater hydrofracture may also increase calving rates, a process that has been strongly implicated in the 2002 collapse of Larsen B Ice Shelf (Scambos et al., 2003). In all simulations of Experiment 2 ice-shelf thickness is allowed to evolve freely. This more realistic experiment permits the evaluation of a more gradual loss of buttressing to the upstream
glaciers and assesses the effect of a dynamic calving front. In all simulations, perturbations to the surface mass balance are ignored as these are expected to be small in comparison to ice dynamic changes resulting from shelf loss (Barrand et al., 2013). Moreover, ocean melting is set to zero in the perturbation experiments unless stated otherwise.

**Table 1.** Complete list of all perturbation experiments including sensitivity simulations as well as grid resolutions for Experiments 1 and 2.

| Name of experiment | Model | Grid resolution [$m$] |
|---|---|---|
| Experiment 1 | BAS-APISM | 900 |
| Experiment 1 | PSU3D | 4000, 2000, 1000 |
| Experiment 1 with default sliding | BISICLES | 4000, 2000, 1000, 500 |
| Experiment 1 with linear Weertman | BISICLES | 1000 |
| Experiment 1 with cubic Weertman | BISICLES | 1000 |
| Experiment 1 with Coulomb sliding | BISICLES | 1000 |
| Experiment 1 with bedrock from Huss and Farinotti (2014) | BISICLES | 1000 |
| Experiment 2 with zero melt | BISICLES | 1000 |
| Experiment 2 with zero melt | PSU3D | 1000 |
| Experiment 2 'moderate' ocean melt | PSU3D | 1000 |
| Experiment 2 'extreme' ocean melt | PSU3D | 1000 |

## 3   Results

### 3.1   Experiment 1: Immediate ice-shelf collapse

Projections of sea-level rise from Larsen C embayment glaciers following immediate shelf collapse (Experiment 1) are small, ranging from 0.5-1.5 mm by 2100 and 0.6-1.6 mm to 2300 (Figure 4a). The sea-level curve rises in the first two decades in response to loss of backstress provided by the shelf, then decelerates with tributary glaciers adjusting to the new configuration ∼25 years after collapse. Grounding-line retreat of >5 km and extensive dynamic thinning (>0.6 m a$^{-1}$, propagating ∼75 km inland) is restricted to five outlet glaciers in the southern part of the embayment (Figure 5). In contrast, immediate collapse of George VI Ice Shelf perturbs upstream grounded tributaries by up to 0.8 m a$^{-1}$ averaged over 300 years, and results in 4-11 mm total sea-level rise by 2300 (Figure 4b). The more dramatic response of George VI tributary glaciers means that they have not yet reached steady-state by 2100 (Figure 4d), and PSU3D simulations continue to contribute to sea level well beyond this date. This discrepancy between the sheet-shelf models may be attributed to a combination of differences in initialisation, inferred basal traction fields, and that PSU3D is not as close to steady-state as BISICLES following initialisation and spin-up (Figures 3 and A4). Moreover, ice-sheet thinning in response to the collapse event propagates further upstream in PSU3D and is more widespread than in BISICLES, leading to higher rates of mass loss despite similar predicted grounding-line retreat

(Tables A1 and A2). Such a response has been previously attributed to differences in the underlying model physics (L1L2, A-HySSA). Using synthetic geometries, A-HySSA models have shown to be more sensitive to grounding-line advance as well as retreat. These differences are most likely caused by the neglecting of vertical shearing terms in the pure membrane ice-sheet models (Pattyn et al., 2013).

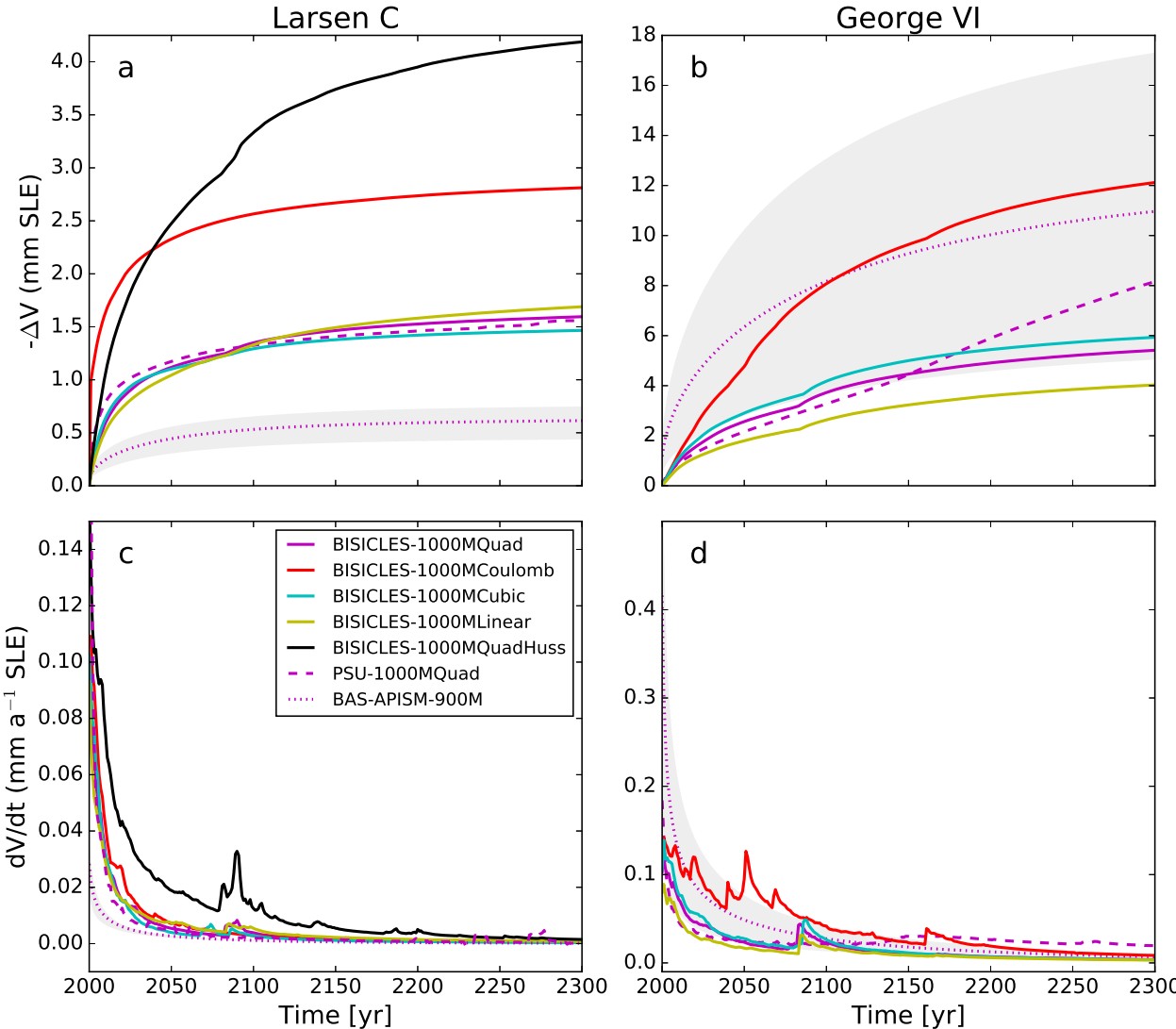

**Figure 4.** Upper panels (a,b) show SLR projections from Experiment 1 (immediate shelf collapse) for BISICLES (solid lines), PSU3D (dashed lines), and BAS-APISM (dotted line). Lower panels (c,d) show the derivative (rate of change) of the corresponding SLR projections in the upper panels (a,b). Grey shading displays uncertainty associated with SLR projections from BAS-APISM. Uncertainties are quantified by a Monte-Carlo simulation (see (Schannwell et al., 2016)). Note the different y-axis scales. Projections with Huss and Farinotti (2014) dataset is only available for Larsen C. Quad=Quadratic

While there is a notable grid dependence in BISICLES projections (Figure A6), this is much reduced in the PSU3D projections supporting the findings of previous modelling studies (Pollard and DeConto, 2012a) that ice-sheet models with the implementation of an internal flux boundary condition are less sensitive to grid resolution. The required first order convergence (Cornford et al., 2016) of the sea-level rise projections in the BISICLES simulations is met for simulations at 1 km and 0.5 km

resolution. To facilitate comparison between the two sheet-shelf models a 1 km grid is employed for Experiment 2.

At 1 km resolution, the combined sea-level rise by 2300 from glaciers in both embayments ranges from 5.6 to 10.4 mm sea-level equivalent, with >60% of the total provided by George VI outlet glaciers. BAS-APISM projects a similar total to 2300 (11.5 mm), though a poor match in simulated spatial patterning of dynamic thinning (Figures 5 and 6) shows that the simplified
model physics and statistical approach to grounding-line retreat do not perform satisfactorily in some areas.

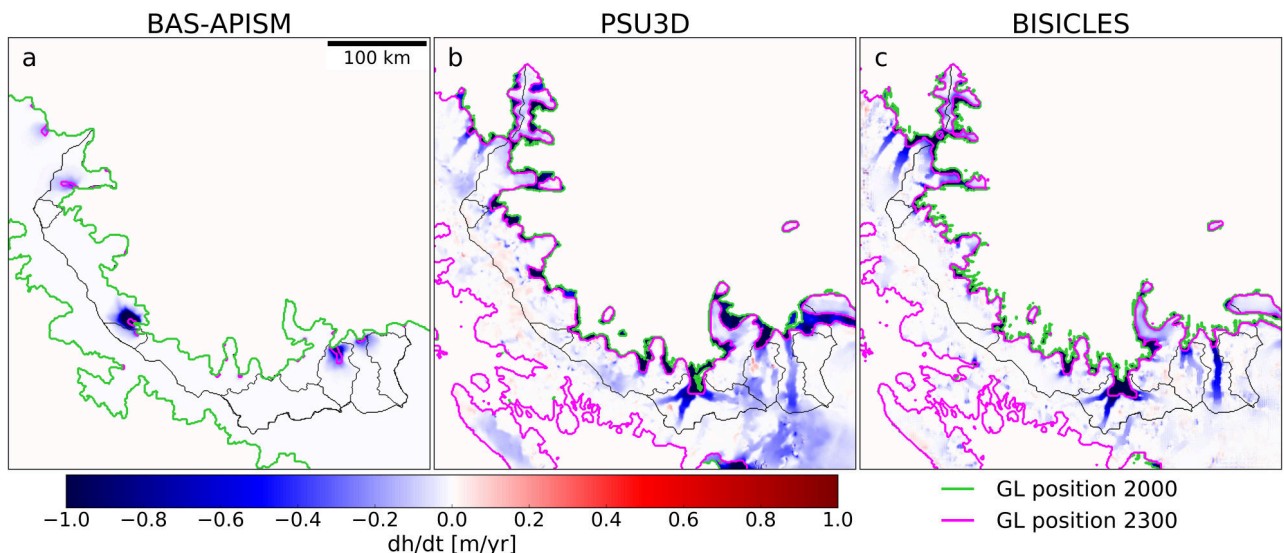

**Figure 5.** Dynamic thinning (dh/dt) pattern from Experiment 1 (immediate shelf collapse) averaged over the simulation period 2000-2300 for the Larsen C embayment in (a) BAS-APISM, (b) PSU3D, and (c) BISICLES. Black lines denote modelled drainage basins.

Across all three ice-sheet models, and in both Larsen C and George VI embayment domains, ice-shelf collapse does not result in widespread and extensive grounding-line retreat (Figures 5 and 6). This was expected for Larsen C outlet glaciers due to a combination of prograde-sloping bedrock topography and the moderate backstress currently provided by the shelf (Fürst
et al., 2016). George VI Ice Shelf, however, provides both strong buttressing (Fürst et al., 2016) and mostly marine-based outlet glaciers on retrograde sloping bedrock topography (Figure 1b), conditions expected to be favourable for marine ice sheet instability. Despite this, grounding-line retreat of George VI outlet glaciers is limited to a few locations and <15 km in length (Figure 6). These findings suggest that stabilising forces such as basal and lateral drag may provide enough resistance for the ice sheet in western Palmer Land to remain in a stable configuration following the initial response to ice-shelf collapse.
This is supported by earlier modelling studies with idealised geometries, showing that the magnitude of grounding-line retreat is a function of the retrograde sloping channel width (Gudmundsson et al., 2012; Gudmundsson, 2013). The smaller the channel width, the less retreat was simulated (Gudmundsson et al., 2012). Considering the small size of the drainage basins in the peninsula region with channel widths <30 km, the remaining lateral buttressing from shear margins likely impedes any runaway grounding-line retreat.

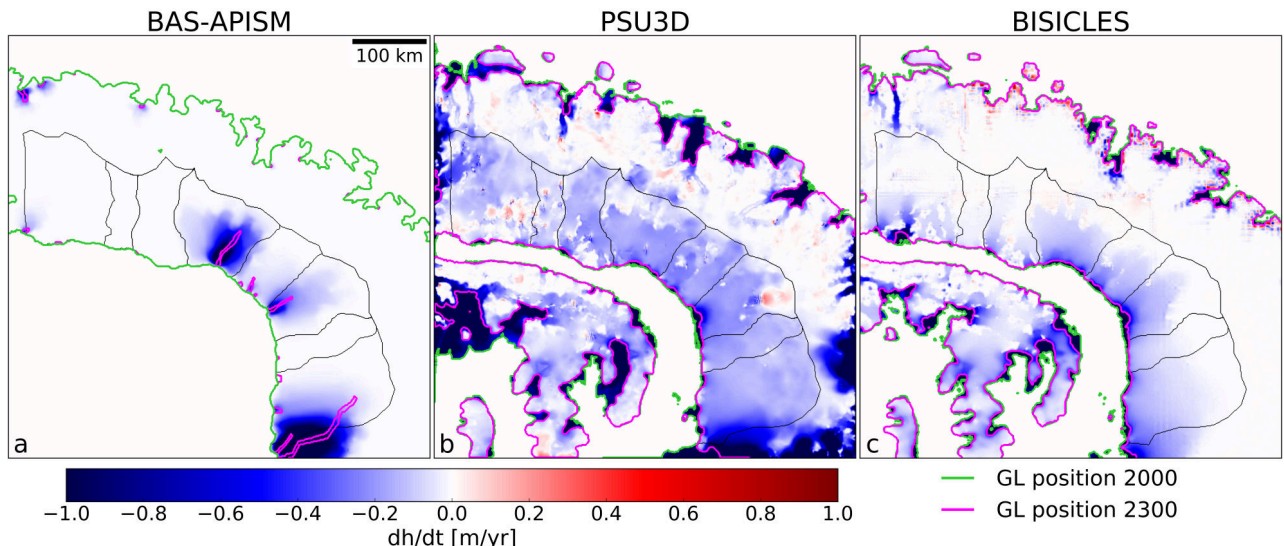

**Figure 6.** Dynamic thinning (dh/dt) pattern from Experiment 1 (immediate shelf collapse) averaged over the simulation period 2000-2300 for the George VI embayment in (a) BAS-APISM, (b) PSU3D, and (c) BISICLES. Black lines denote modelled drainage basins.

## 3.2 Experiment 2: Gradual ice-shelf retreat

When ice-shelf frontal changes are explicitly simulated (Experiment 2) with sheet-shelf models (PSU3D, BISICLES) using a stress-field dependent calving law, sea-level rise projections span a much larger range. With forcing from the Representative Concentration Pathway (RCP) 8.5 'high emission' scenario, Larsen C and George VI embayment basins combined provide up to 23 mm sea-level equivalent ice loss by 2300 (Figure 7b), with 95% of this total coming from George VI tributary glaciers. The contribution to the sea-level budget from Larsen C embayment glaciers is small (<1.5 mm) and remains equivalent to Experiment 1. The sea-level commitment from Larsen C glaciers is modest as complete shelf collapse is not forecast until 2150 in RCP8.5 (Figure 7c), and only 45-60% of the shelf area is lost by 2300 in RCP4.5 ('business-as-usual' scenario). This leads to limited grounding-line retreat and dynamic thinning is restricted to five outlet glaciers in the southern part of the embayment (Figure 8a). The larger grounded area loss simulated with PSU3D for Larsen C (Figure 7c) is not a response to loss of buttressing force, rather it is due to a more seaward advanced initial grounding-line position introduced in the model spin-up phase, the effect of which on sea-level projections is small (0.28 mm sea-level equivalent).

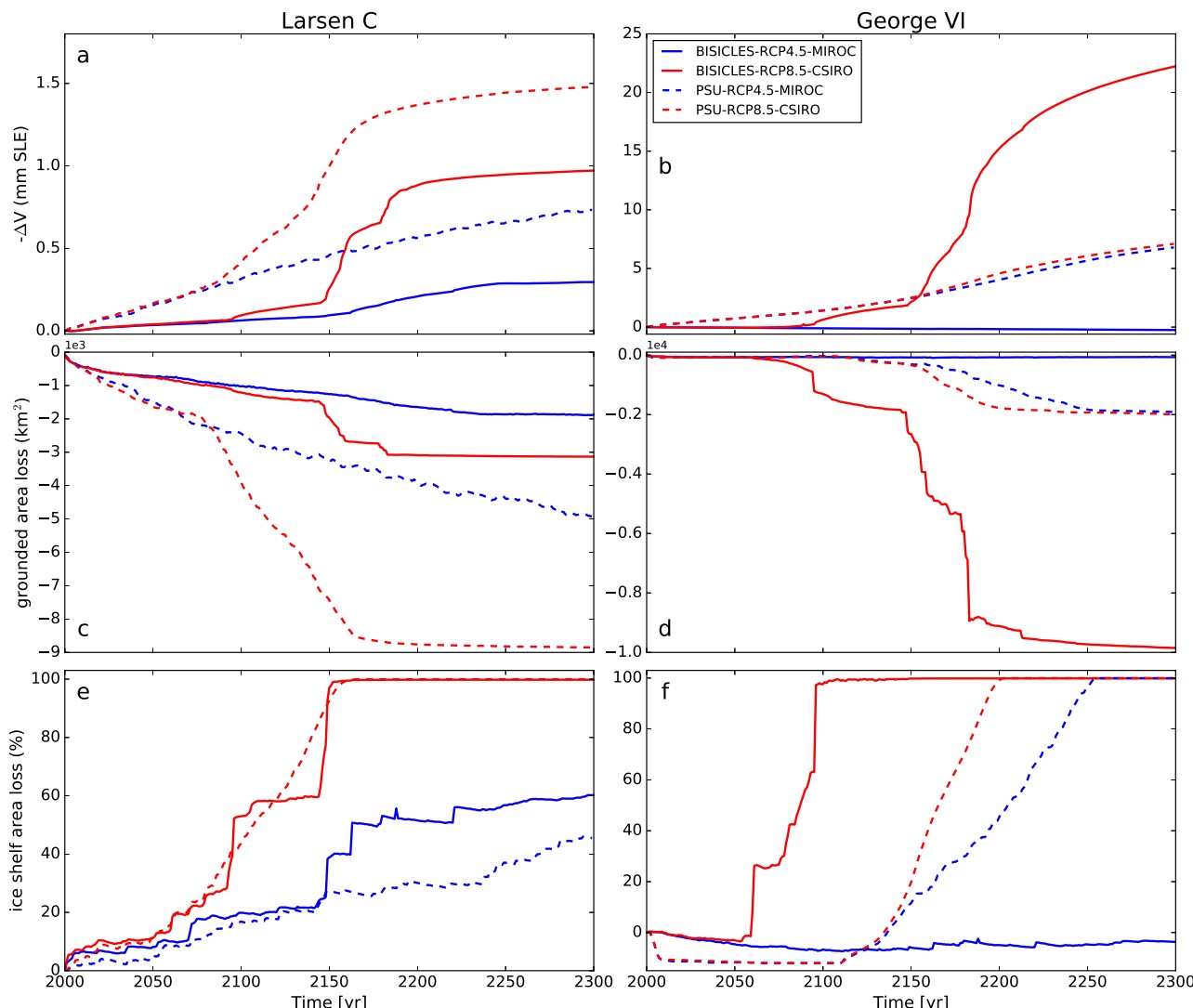

**Figure 7.** SLR projections from Experiment 2 (dynamic calving front) for Larsen C (a) and George VI ice shelves (b) with corresponding area loss of grounded ice (c,d) and ice shelf area loss (e,f). MIROC and CSIRO denote selected global climate model forcing.

Although projections for Larsen C Ice Shelf glaciers agree reasonably well in absolute numbers across both sheet-shelf models despite differences in their underlying physics, projections diverge for simulations of George VI Ice Shelf glaciers (Figure 7b). Experiment 2 (dynamic calving front) simulations with PSU3D provide very similar sea-level projections to Experiment 1 runs for George VI (immediate ice-shelf collapse; 6.8-7.1 mm from RCP4.5 and 8.5, respectively). In contrast, BISICLES projects little sea-level rise under RCP4.5 for George VI, as the amount of meltwater available for hydrofracturing is insufficient to initiate ice-shelf collapse or retreat due to the different implementation of the calving law (See Section 2.2). Under RCP8.5, break-up of George VI Ice Shelf occurs at approximately 2100 (Figure 7d), resulting in widespread grounding-

line retreat and sea-level rise of 22 mm by 2300. Both sheet-shelf models project similar sea-level rise up to 2150 in Experiment 2 for the George VI domain. In the BISICLES RCP8.5 simulation, however, following the collapse of the shelf, calving fronts and grounding lines retreat further back into the marine-based sectors (Figure 8d). After ice-shelf collapse, grounding line and calving front for all drainage basins are almost in identical locations. Increasing rates of calving permit the grounding line to

5 retreat much further inland in the RCP8.5 BISICLES simulation for George VI. As this enhanced grounding-line retreat is only present in Experiment 2, it suggests that this retreat is most likely due to a combination of the dynamic calving front and the marine ice-sheet instability mechanism. Even with a dynamic calving front, enhanced grounding-line retreat for George VI is not triggered before some time after ice-shelf collapse (>15 years, Tables A3 and A4), indicating that fast grounding-line retreat is not triggered before the calving front along with the grounding line reaches a retrograde sloping bedrock topography.

As a result of widespread grounding-line retreat for George VI in the RCP8.5 scenario (Figure A5), extensive dynamic thinning occurs (>1 m a$^{-1}$), extending up to 100 km inland in the southern parts of the embayment (Figure 8d). PSU3D simulations do not show enhanced grounding-line retreat in this sector.

The discrepancy in sea-level rise projections between Experiment 1 and Experiment 2 is a result of the different applied perturbations. In Experiment 1, the entire ice shelf is removed at the start of the simulation before a fixed calving front is employed.

In contrast, Experiment 2 with the crevasse calving law has much more potential to vary. Our simulations show that either very little ice can calve (Figure 7, RCP4.5 scenario) or given enough surface water the entire shelf can collapse and emerging new floating areas that were formerly grounded keep on calving. So unlike Experiment 1 where collapse is only enforced once, repeated/continuing collapse of the shelf can occur in Experiment 2 (Figure 7, RCP8.5 BISICLES simulation).

We attribute the good agreement across both models for Larsen C to the fact that the area of the marine-based sectors is limited

in this domain (2.1 mm contained in marine-based sectors) due to the very mountainous bedrock topography constraining potential grounding-line retreat. This is supported by all simulations across all ice-sheet models as even under a wide range of different forcings the Larsen C embayment does not contribute more than 4.2 mm by 2300. The greater potential to initiate grounding-line retreat is presented by George VI Ice Shelf where much of the ice sheet is marine based with retrograde sloping bedrock topography (Figure 1b). As this large grounding-line retreat is only initiated in the BISICLES simulation, large differ-

ences in sea-level rise projections occur. The most likely explanation for this differing behaviour is due to the difference in the inferred basal traction coefficient fields that affects each model's response to ice-shelf removal. PSU3D predicts much higher-friction bedrock conditions in the George VI embayment than BISICLES (Figure 2). These high friction bedrock conditions result in little acceleration of the major outlet glaciers following ice-shelf breakup. This in turn means that the calving law applied to only floating ice cells cannot drive the initial retreat into the marine based sectors as the outlet glaciers do not thin

sufficiently to form floating ice tongues. In contrast in the RCP8.5 BISICLES simulation for George VI, speed-up in response to ice-shelf breakup leads to enhanced dynamic thinning of the main outlet glaciers. This thinning in conjunction with the calving law drives the calving front into the marine-based sectors where further retreat is initiated by a combination of the marine ice-sheet instability and the meltwater driven calving law, resulting in the simulated much higher sea-level rise projections.

### 3.3 Uncertainty assessment

A range of sensitivity experiments were undertaken to assess the robustness of our model simulations to additional forcings. To assess the impact of an additional ocean forcing, a pair of basal melt anomalies were applied to areas of fully-floating ice in addition to the freely-evolving calving front forcing (Experiment 2). In a first, 'moderate' simulation, the anomaly was set to the current thinning signal of the respective ice shelf (Paolo et al., 2015) for the duration of the forecast period (0.5 m a$^{-1}$ for Larsen C, 1.1 m a$^{-1}$ for George VI). In a second, 'extreme' scenario, the same initial anomaly was applied, then increasing linearly to 3 times the current thinning signal by 2100, remaining at this magnitude to 2300. In each case, sea-level projections with these additional forcings are within 0.2 mm sea-level equivalent of simulations without additional forcings: in other words, it is ice-shelf break-up in combination with the calving criteria that dominates our results. As the basal boundary condition remains poorly constrained in ice-sheet models, yet our model projections show a strong dependence on this condition, Experiment 1 (immediate shelf collapse) was repeated with BISICLES using a range of basal sliding laws. Each of the traditionally-employed power laws result in similar sea-level rise projections to 2300 (1.4-1.6 mm for Larsen C embayment glaciers, and 4-6 mm for George VI glaciers, respectively; Figure 4). Projections to 2300 increase by a factor of two for simulations using a Coulomb-limited sliding law (Tsai et al., 2015), resulting in ∼3 mm from Larsen C glaciers and ∼12 mm for George VI glaciers. This type of basal sliding law reduces the basal drag in a mobile ∼1 km layer which forms immediately upstream of the grounding line, resulting in greater discharge throughout the simulations (Tables A1 and A2).

The importance of better constrained boundary conditions in the peninsula region (bedrock topography and ice thickness) is highlighted by a discrepancy between sea-level rise projections for Larsen C embayment basins using different data products. Although total ice volume and ice volume below sea-level differences between ALBMAP and BEDMAP2 products are small (<15%), a more recent higher-resolution dataset (Huss and Farinotti, 2014) provides an increase of ∼100% in ice volume below sea level. When incorporated into ice-sheet model simulations, this altered bedrock topography results in larger grounding-line retreat rates for some basins occupying deeper bedrock troughs. A consequence is a sea-level rise projection for Experiment 1 (immediate ice-shelf collapse) with the reference sliding law (Equation 2) that increases by a factor of ∼3 (4.2 mm) for the Huss and Farinotti (2014) bedrock topography dataset, underlining the significance of accurate boundary dataset for sea-level rise projections.

In addition, our experiments show that for simulations of grounding-line motion in response to ice-shelf breakup sheet-shelf models are necessary. The simple model BAS-APISM fails to reproduce the results of the sheet-shelf models due to the simplified physics. Even across sheet-shelf models differences in model physics, model initialisation, calving law implementation and other numerics (e.g. meshing) can lead to substantially different projections under the same forcing (Figure A5). Sea-level rise projections are most sensitive to the choice of sliding law and bedrock geometry. The peninsula is not the only region where these parameters highly affect decadal to centennial sea-level rise projections as similar conclusions were drawn from modelling of outlet glaciers in the Amundsen Sea embayment (Nias et al., 2018). The wide range of sea-level rise responses to different forcing parameters underlines the need for perturbed ensembles to explore key parameter uncertainties (e.g. basal sliding law) for sea-level rise projections in greater detail for the peninsula region. Owing to the increase in computer power

these type of ensemble projections have become feasible at the regional (e.g. Nias et al., 2016) and continental scale (e.g. DeConto and Pollard, 2016).

## 3.4 Comparison with Larsen B Ice Shelf collapse response

To further assess the impact of ice-shelf break-up, five drainage basins from the Larsen C embayment (LarI-LarV, Figure 8) and George VI embayment (GeoI-GeoV, Figure 8) were selected for additional analysis. This provides a comparison to real-world examples of the magnitudes and pattern of glacier response to ice-shelf collapse. For Experiment 1 (immediate ice-shelf collapse), the speed up following ice-shelf removal is short lived ($\sim$15 yrs) for both models. Maximum speed-up of $\sim$300% is possible, though the mean maximum speed-up is $\sim$50% (Tables A1 to A4). These values are smaller than those observed following Larsen B collapse with a maximum of 8 fold speed up (Rignot et al., 2004). This may be due to the different areas selected for the speed up calculation. Both rates of ice discharge (mass loss) and grounding-line retreat are greatest immediately following shelf collapse. For 65% of the selected 10 drainage basins, more than 50% of the total modelled grounding-line retreat takes place within 15 years of ice-shelf collapse. Maximum mass loss rates for Larsen C (1.6-5.1 Gt a$^{-1}$) are smaller than observations for a similar time period for Larsen B (8.0 Gt a$^{-1}$) (Scambos et al., 2014). Responses of individual drainage basins for Larsen C and George VI are highly variable with grounding-line retreat ranging from 3.7 km to 26.8 km for Larsen C and 3.1 km to 12.2 km for George VI (Tables A1 and A2). This high spatial variability in response across the selected basins indicates that the importance of ice-shelf buttressing is also highly spatially variable. Most of the grounding-line retreat, in particular for Larsen C, occurs in areas of bedrock channels (Figure 5). Since these deep bedrock channels are smaller for Larsen C, this leads to smaller mass loss than for George VI even though maximum grounding-line retreat numbers are larger. But grounding-line retreat is spread across a wider area of the drainage basin front at George VI (Figure 6).

For Experiment 2, maximum grounding-line retreat (4 km to 33.4 km, Tables A3 and A4) is of similar magnitude to Experiment 1 (3.1 km to 26.8 km, Tables A1 and A2) including the spatial variability across the selected basins for PSU3D in both embayments. However, for BISICLES this holds only true for the Larsen C embayment where mass loss averaged across the five basins remains at 0.1 $Gt\ a^{-1}$ for both experiments. For the George VI domain the spatial variability across the basins is strongly reduced with maximum grounding line retreat now ranging from 10 km to 25.5 km (Table A4). When averaged over the five basins grounding-line retreat increases from 6.4 km for Experiment 1 to 21.3 km in Experiment 2 for George VI. The retreat in this experiment spreads over the entire width of the drainage basin front (Figure 8d), resulting in an increase of mass loss over the 300 years from 0.5 $Gt\ a^{-1}$ in Experiment 1 to 3.0 $Gt\ a^{-1}$ in Experiment 2. This is in agreement with computed sea-level rise projections (Figure 7). Significant speed-up is absent in the years following ice-shelf removal across all basins due to the more gradual loss of buttressing in Experiment 2 (compared to the complete ice-shelf removal in Experiment 1). This results in a less dramatic dynamic response than in Experiment 1, with the exception of several basins of George VI Ice Shelf where retreat rates can lead to large mass losses. The gradual loss of buttressing simulated by Experiment 2 leads to grounding-line retreat and mass loss response occurring >15 years after ice-shelf removal.

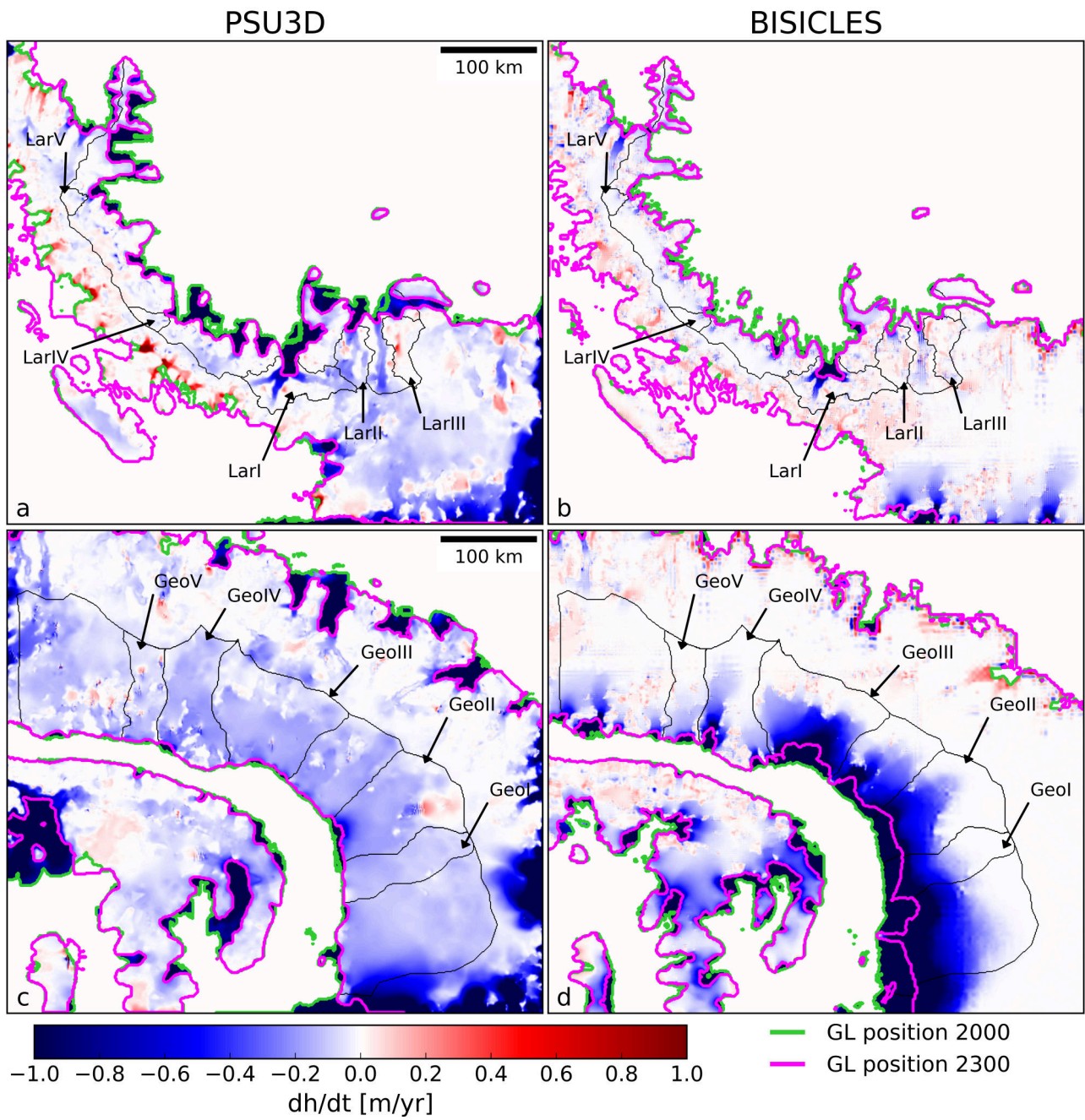

**Figure 8.** Dynamic thinning (dh/dt pattern from Experiment 2 (dynamic calving front) averaged over the simulation period 2000-2300 in the RCP8.5 scenario for Larsen C (a,c) and George VI embayments (b,d). Black lines denote modelled drainage basins. LarI-V and GeoI-V indicate drainage basins selected for analysis.

## 4 Conclusions

The most important contributor to the global sea-level budget to 2300 from Antarctic Peninsula ice-shelf-ice sheet dynamics are glaciers in western Palmer Land feeding George VI Ice Shelf. Our envelope of sea-level rise projections ranges from 4-12 mm sea-level equivalent water by 2300 in Experiment 1, to 6-22 mm sea-level equivalent water by 2300 in Experiment 2

for George VI. As the highest projection represents only 55% of the grounded ice below sea level in this region (Fretwell et al., 2013), there may yet be even more ice at risk to dynamic mass loss. This is in contrast to the Larsen C embayment where the majority of the projections result in loss of more than 55% of grounded ice below sea-level. The highest projections remove all grounded ice below sea level as well as ice elsewhere in the basin. The sea-level rise induced by ice-shelf removal in the Larsen C embayment is therefore largely limited by the small area of marine-based sectors. All projections are relatively insensitive

to increased ocean forcing, yet are highly sensitive to changes in the basal boundary condition and the choice of boundary data set, highlighting the need for improved bed topography data and a more rigorous uncertainty analysis. While Larsen C Ice Shelf's recent calving event may increase its vulnerability to ice-shelf instability, our simulations under a wide range of future forcing scenarios show that the sea-level commitment of Larsen C embayment glaciers following shelf collapse or retreat are limited to less than 4.2 mm by 2300 (0.6-4.2 mm for Experiment 1; 0.4-1.5 mm for Experiment 2). Individual drainage basin

analysis indicates a wide range of responses in response to ice-shelf removal, but overall ice flow speed and mass changes are expected to be of similar magnitude to those observed following the 2002 collapse of Larsen B Ice Shelf.

## Appendix A

*Competing interests.* The authors declare that there are no competing interests.

*Acknowledgements.* C.S. was supported by a PhD studentship from the University of Birmingham. The computations described in this paper

were performed using the University of Birmingham BlueBEAR HPC service, which provides a High Performance Computing service to the University's research community. See http://www.birmingham.ac.uk/bear for more details. We acknowledge the World Climate Research Programmes Working Group on Coupled Modelling, which is responsible for CMIP, and we thank the climate modeling groups for producing and making available their model output (available at http://pcmdi9.llnl.gov/). For CMIP the U.S. Department of Energy's Program for Climate Model Diagnosis and Intercomparison provides coordinating support and led development of software infrastructure in partnership

with the Global Organization for Earth System Science Portals. We thank the editor Olivier Gagliardini, Lionel Favier and an anonymous reviewer for comments which improved the manuscript.

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

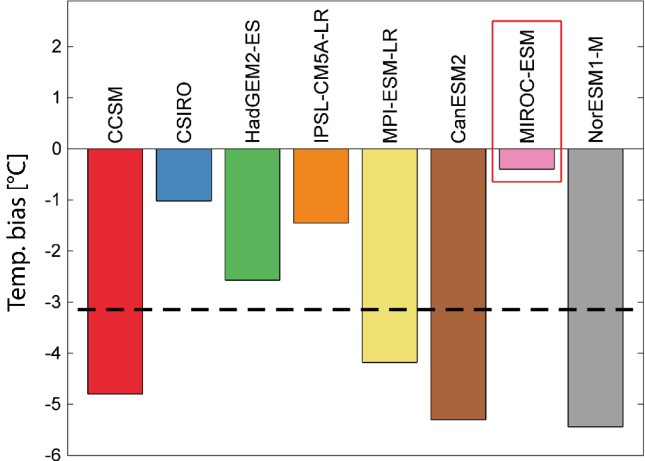

**Figure A1.** Near-surface temperature bias for the baseline period 1980-2005 in GCMs for RCP4.5 projections in relation to ERA-Interim. Dashed black line indicates multi-model mean (-3.1±2.0°C). The selected forcing is highlighted by the red box.

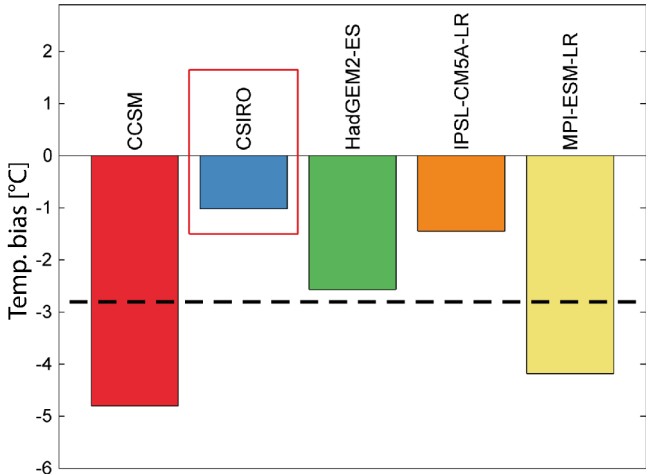

**Figure A2.** Near-surface temperature bias for the baseline period 1980-2005 in GCMs for RCP8.5 projections in relation to ERA-Interim. Dashed black line indicates multi-model mean (-2.8±1.7°C). The selected forcing is highlighted by the red box.

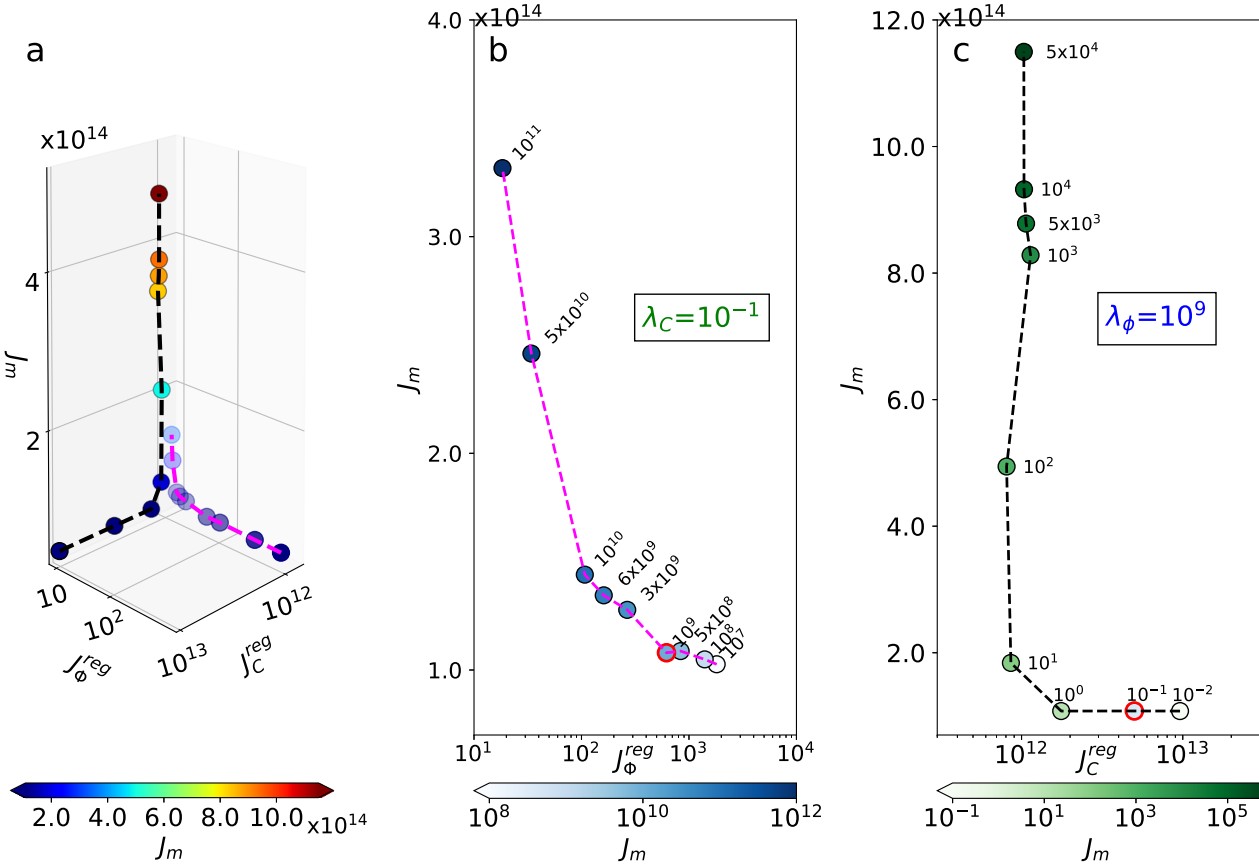

**Figure A3.** BISICLES L-curve analysis to select Tikhonov parameters $\lambda_\phi$ and $\lambda_C$: (a) 3-D scatter plot of the model-data misfit $J_m$ as a function of the regularisation terms $J_C^{reg}$ and $J_\phi^{reg}$. (b) 2-D cross section for variable $\lambda_\phi$ and $\lambda_C$ fixed at $10^{-1}$ Pa$^{-2}$ m$^6$ a$^{-4}$. (c) Reverse case with constant $\lambda_\phi$ at $10^9$ m$^4$ a$^{-2}$ and $\lambda_C$ varying. The units of $J_m$ and $J_C^{reg}$ are m$^4$ a$^{-2}$ and Pa$^2$ m$^{-2}$ a$^2$, respectively. $J_\phi^{reg}$ is unitless. Selected values are highlighted by red circles in (b) and (c). The layout was inspired by Berger et al. (2016).

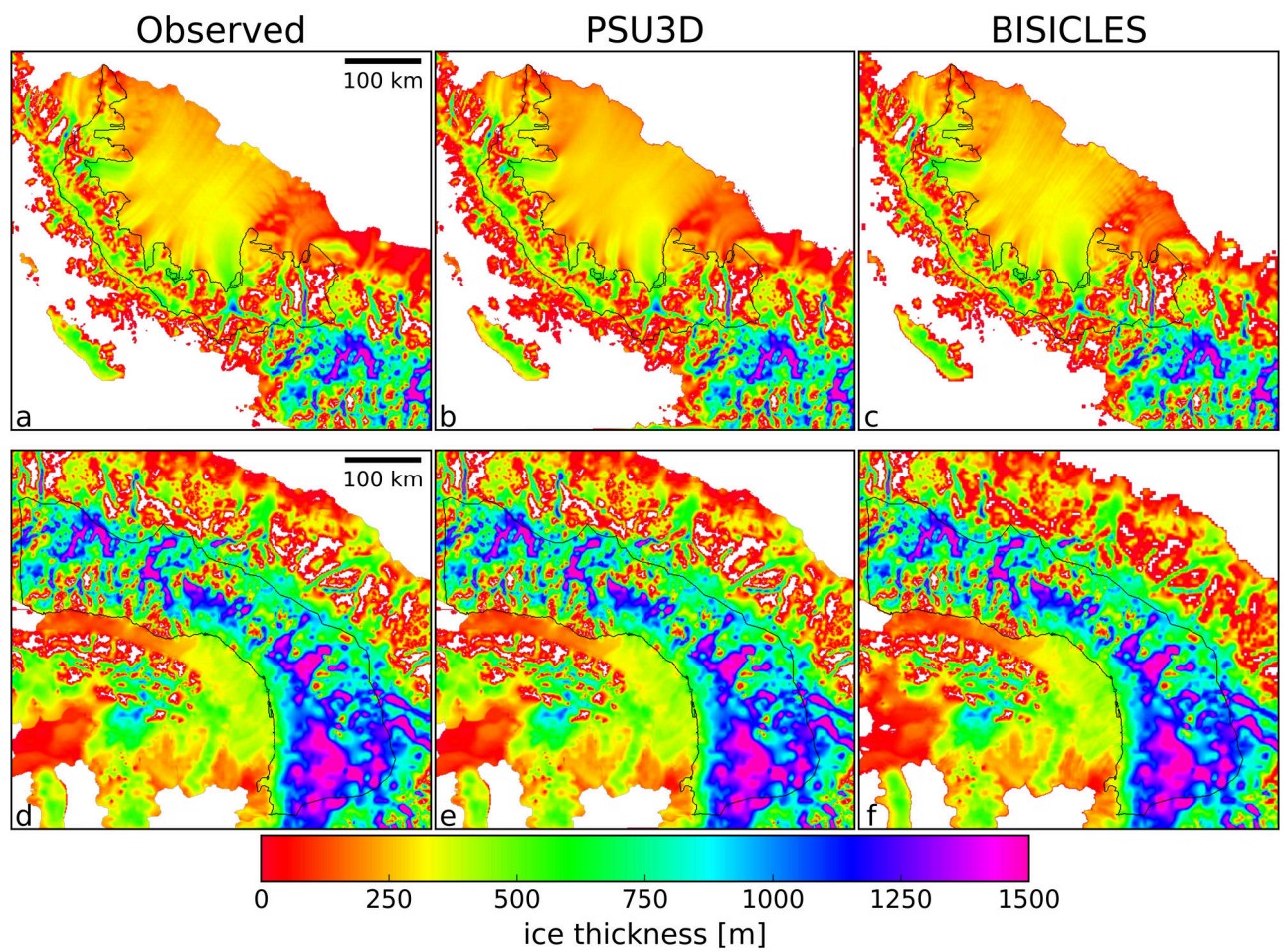

**Figure A4.** Observed ice thickness distribution (BEDMAP2(Fretwell et al., 2013)) for Larsen C (a) and George VI (d) embayments and modelled ice thickness distribution after spin-up for Larsen C (b,c) and George VI (e,f) embayments. Black lines denote modelled drainage basins.

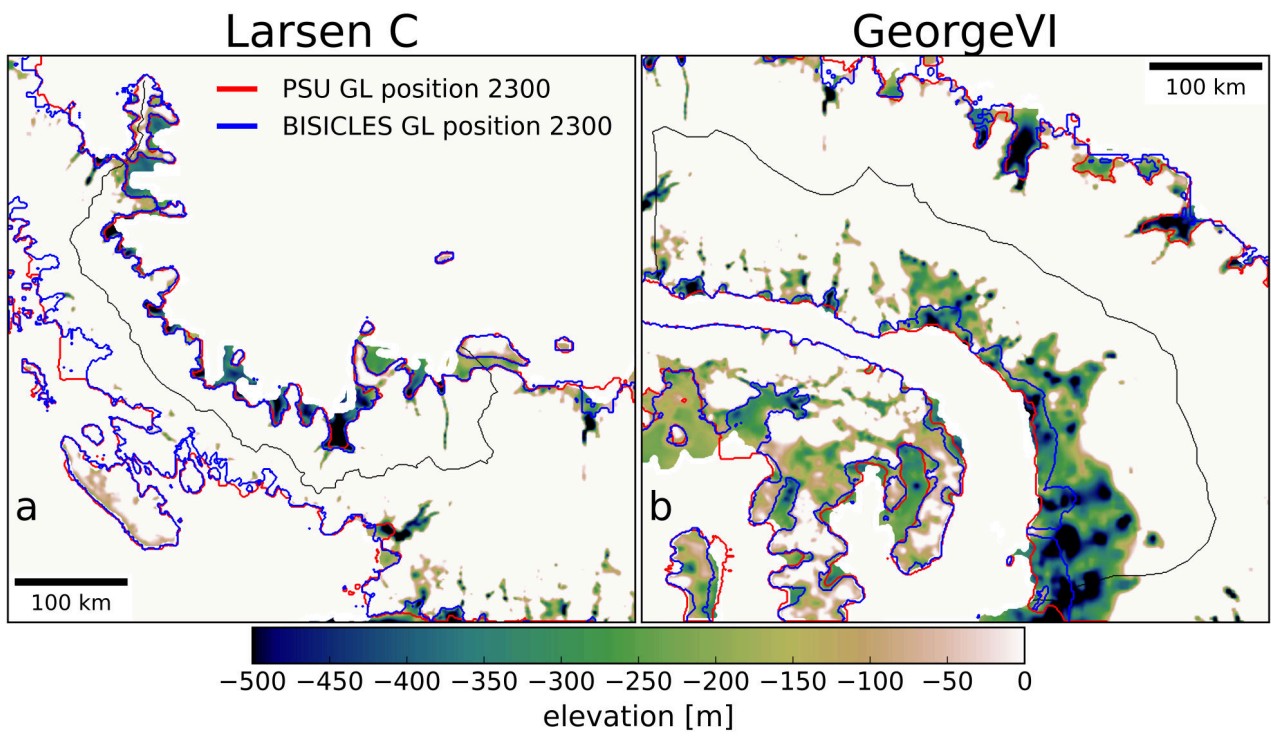

**Figure A5.** Comparison of modelled grounding-line positions from Experiment 2 (dynamic calving front) for RCP8.5 scenario for Larsen C (a) and George VI embayments (b). Black lines denote modelled drainage basins.

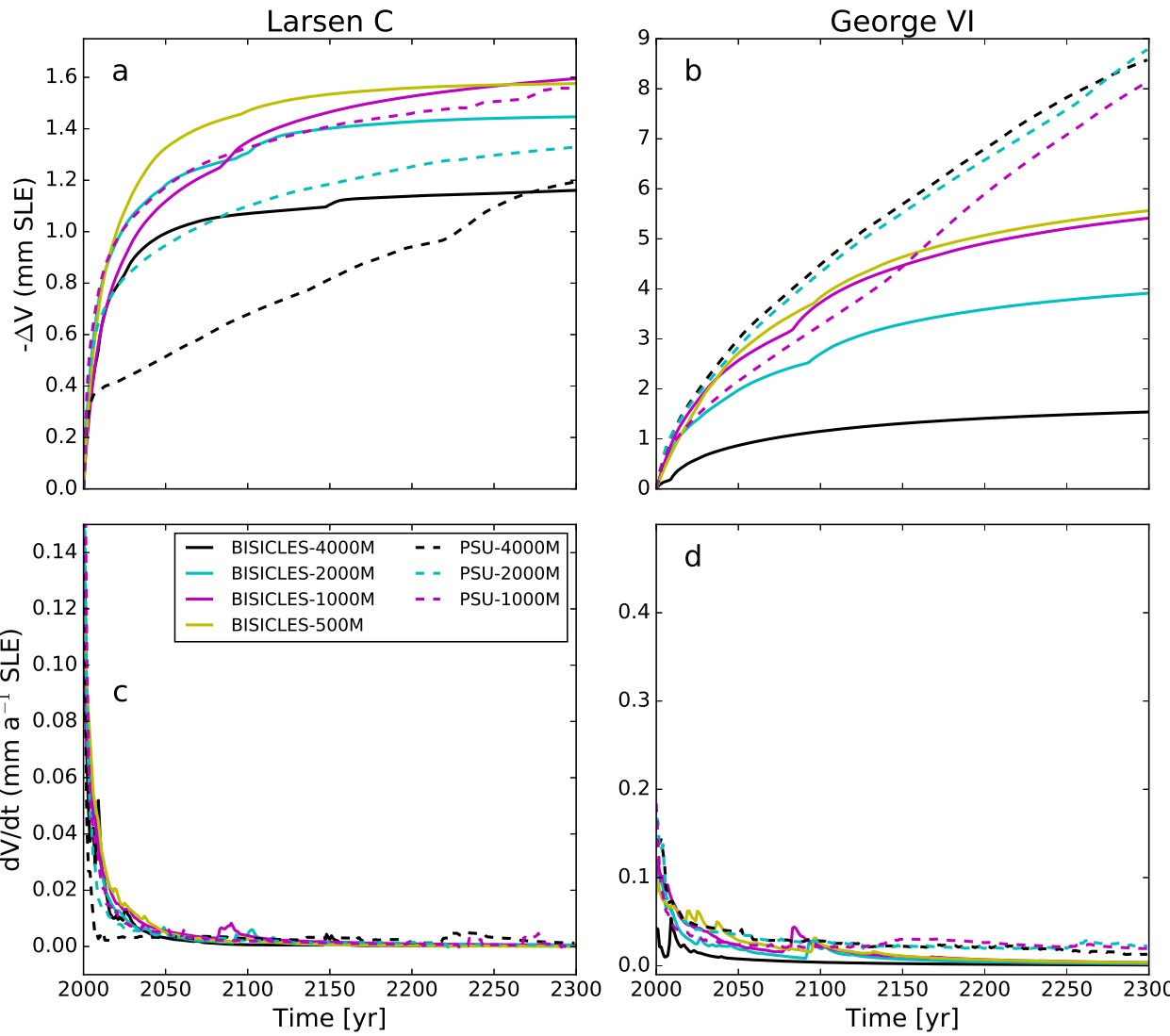

**Figure A6.** Upper panels (a,b) show sea-level rise projections from Experiment 1 (immediate shelf collapse) at different horizontal resolutions for BISICLES (solid lines) and PSU3D (dashed lines). Lower panels (c,d) show the derivative (rate of change) of the corresponding sea-level rise projections in the upper panels (a,b). Note the different y-axis scales.

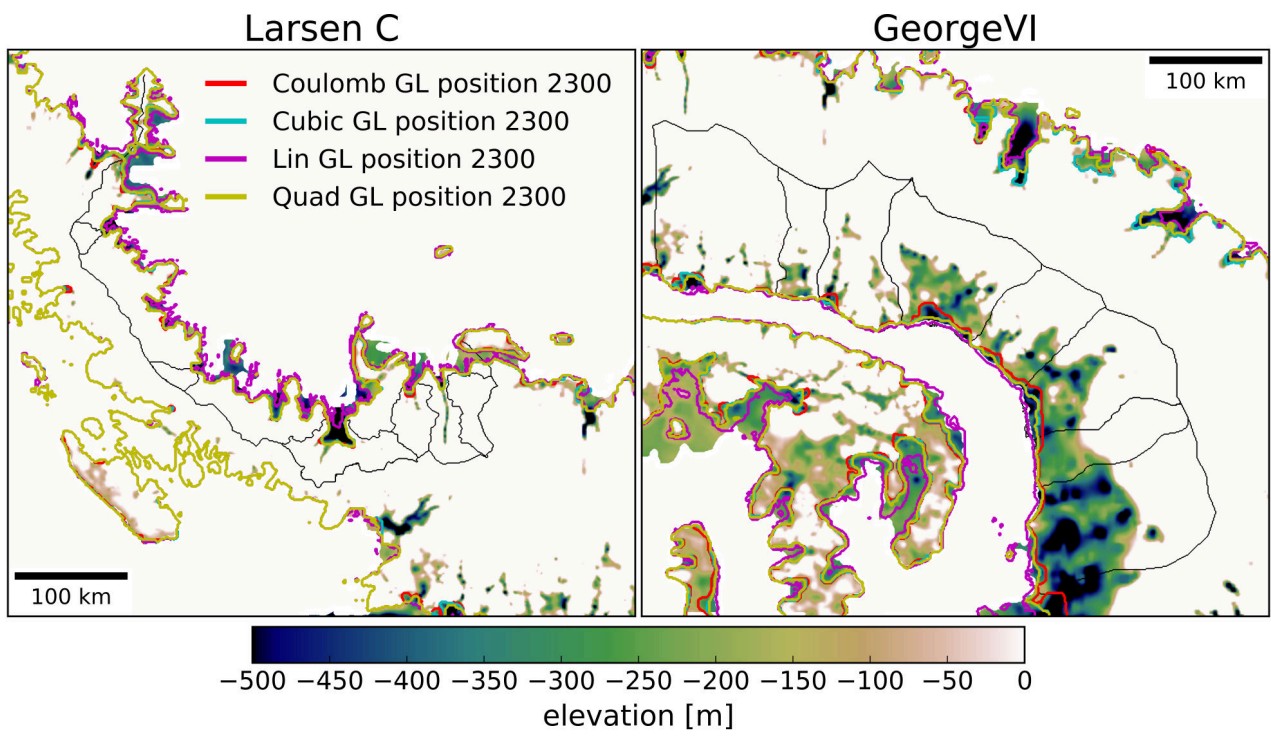

**Figure A7.** Comparison of modelled grounding-line positions using BISICLES with different basal sliding laws for Larsen C (a) and George VI embayments (b). Black lines denote modelled drainage basins.

**Table A1.** Maximum grounding-line retreat (dGL [$km$]), mass change rate (dM/dt [$Gt\ a^{-1}$]) and speed up ($dU/dt_{15}$) for selected sample basins in Larsen C embayment for Experiment 1 (immediate collapse). Subscript indicates that numbers are averaged over 15 years after ice-shelf collapse with the exception of $dGL_{15}$ where it is the sum over this period. For columns without subscript the numbers are for the entire 300 year simulation period. For speed up, in addition to the average number over 15 years ($dU/dt_{15}$), the maximum speed up in this time period is provided ($dU/dt_{max}$). Speed up was calculated for a region within five kilometers of the current grounding line. As BAS-APISM does not simulate the ice shelf, speed up calculations were not carried out.

| | BAS-APISM | | PSU3D | | | BISICLES | | | BISICLES-Coulomb | | |
|---|---|---|---|---|---|---|---|---|---|---|---|
| | dGL | dM/dt | dGL | dM/dt | $dU/dt_{15}$ | dGL | dM/dt | $dU/dt_{15}$ | dGL | dM/dt | $dU/dt_{15}$ |
| | ($dGL_{15}$) | ($dM/dt_{15}$) | ($dGL_{15}$) | ($dM/dt_{15}$) | ($dU/dt_{max}$) | ($dGL_{15}$) | ($dM/dt_{15}$) | ($dU/dt_{max}$) | ($dGL_{15}$) | ($dM/dt_{15}$) | ($dU/dt_{max}$) |
| LarI | 0.0 | 0.0 | 26.8 | 0.3 | 1.0 | 21.4 | 0.5 | 1.4 | 14.1 | 0.5 | 0.7 |
| | (-) | (0.0) | (21.2) | (3.6) | (1.8) | (15.2) | (5.1) | (2.0) | (13.7) | (6.4) | (1.1) |
| LarII | 6.3 | 0.2 | 21.0 | 0.2 | 0.7 | 7.1 | 0.1 | 1.4 | 6.8 | 0.2 | 0.8 |
| | (-) | (2.2) | (11.2) | (0.9) | (1.0) | (2.3) | (0.7) | (1.7) | (3.4) | (1.0) | (1.0) |
| LarIII | 1.0 | 0.1 | 9.7 | 0.0 | 0.6 | 3.7 | 0.0 | 1.0 | 0.0 | 0.0 | 1.0 |
| | (-) | (0.9) | (5.0) | (0.2) | (1.0) | (0.0) | (0.0) | (1.2) | (0.0) | (0.0) | (1.0) |
| LarIV | 1.0 | 0.0 | 5.1 | 0.0 | 1.4 | 4.3 | 0.0 | 1.2 | 9.3 | 0.0 | 1.2 |
| | (-) | (0.2) | (1.3) | (0.0) | (1.5) | (0.0) | (0.1) | (1.3) | (1.0) | (0.1) | (1.3) |
| LarV | 0.0 | 0.0 | 10.0 | 0.0 | 2.0 | 4.2 | 0.0 | 1.0 | 3.6 | 0.0 | 1.1 |
| | (-) | (0.0) | (4.7) | (0.0) | (2.9) | (1.1) | (0.0) | (1.3) | (1.0) | (0.0) | (1.2) |

**Table A2.** Same as for Table A1 but for George VI embayment and Experiment 1

| | BAS-APISM | | PSU3D | | | BISICLES | | | BISICLES-Coulomb | | |
|---|---|---|---|---|---|---|---|---|---|---|---|
| | dGL | dM/dt | dGL | dM/dt | $dU/dt_{15}$ | dGL | dM/dt | $dU/dt_{15}$ | dGL | dM/dt | $dU/dt_{15}$ |
| | $(dGL_{15})$ | $(dM/dt_{15})$ | $(dGL_{15})$ | $(dM/dt_{15})$ | $(dU/dt_{max})$ | $(dGL_{15})$ | $(dM/dt_{15})$ | $(dU/dt_{max})$ | $(dGL_{15})$ | $(dM/dt_{15})$ | $(dU/dt_{max})$ |
| GeoI | 0.0 | 0.0 | 5.5 | 0.7 | 0.9 | 5.6 | 0.3 | 1.2 | 3.1 | 0.5 | 1.2 |
| | (-) | (0.0) | (2.1) | (1.8) | (1.6) | (4.0) | (1.6) | (1.3) | (3.0) | (1.9) | (1.3) |
| GeoII | 10.5 | 1.0 | 12.2 | 1.4 | 1.0 | 10.9 | 0.9 | 1.3 | 15.8 | 1.9 | 1.3 |
| | (-) | (7.2) | (6.4) | (4.6) | (2.0) | (6.4) | (4.2) | (1.4) | (7.9) | (5.1) | (1.4) |
| GeoIII | 20.8 | 2.3 | 7.1 | 1.4 | 0.8 | 8.0 | 0.8 | 1.3 | 21.9 | 2.3 | 1.3 |
| | (-) | (17.5) | (5.7) | (5.3) | (1.7) | (7.0) | (4.0) | (1.4) | (9.2) | (6.0) | (1.4) |
| GeoIV | 0.0 | 0.0 | 8.5 | 1.0 | 1.3 | 7.2 | 0.3 | 1.2 | 21.0 | 0.7 | 1.2 |
| | (-) | (0.0) | (5.6) | (3.2) | (1.7) | (4.9) | (1.2) | (1.4) | (5.1) | (1.7) | (1.3) |
| GeoV | 0.0 | 0.0 | 3.1 | 0.3 | 1.1 | 1.4 | 0.0 | 1.1 | 9.2 | 0.1 | 1.1 |
| | (-) | (0.0) | (2.0) | (0.9) | (1.5) | (1.4) | (0.3) | (1.2) | (1.1) | (0.3) | (1.2) |

**Table A3.** Same as for Table A1 but for Experiment 2.

| | PSU3D | | | BISICLES | | |
|---|---|---|---|---|---|---|
| | dGL ($dGL_{15}$) | dM/dt ($dM/dt_{15}$) | $dU/dt_{15}$ ($dU/dt_{max}$) | dGL ($dGL_{15}$) | dM/dt ($dM/dt_{15}$) | $dU/dt_{15}$ ($dU/dt_{max}$) |
| LarI | 29.4 (7.0) | 0.3 (1.0) | 1.0 (1.0) | 19.6 (3.6) | 0.5 (2.0) | 1.0 (1.3) |
| LarII | 33.4 (6.0) | 0.1 (0.2) | 1.0 (1.0) | 2.9 (1.3) | 0.1 (0.4) | 1.1 (1.3) |
| LarIII | 19.3 (4.6) | 0.1 (0.2) | 0.6 (1.0) | 1.5 (0.0) | 0.0 (0.0) | 1.0 (1.0) |
| LarIV | 10.4 (3.8) | 0.0 (0.0) | 1.0 (1.1) | 10.7 (0.0) | 0.0 (0.1) | 1.0 (1.1) |
| LarV | 14.6 (3.1) | 0.0 (0.0) | 1.0 (1.0) | 3.2 (0.0) | 0.0 (0.0) | 1.1 (1.2) |

**Table A4.** Same as for Table A1 but for George VI embayment and Experiment 2.

| | PSU3D | | | BISICLES | | |
|---|---|---|---|---|---|---|
| | dGL ($dGL_{15}$) | dM/dt ($dM/dt_{15}$) | $dU/dt_{15}$ ($dU/dt_{max}$) | dGL ($dGL_{15}$) | dM/dt ($dM/dt_{15}$) | $dU/dt_{15}$ ($dU/dt_{max}$) |
| GeoI | 4.7 (0.0) | 0.5 (0.6) | 1.0 (1.0) | 22.3 (1.2) | 3.3 (0.5) | 1.0 (1.0) |
| GeoII | 11.5 (1.1) | 1.5 (2.1) | 1.0 (1.0) | 23.5 (2.7) | 4.2 (1.7) | 0.9 (1.0) |
| GeoIII | 5.1 (0.5) | 1.1 (1.6) | 1.0 (1.0) | 25.3 (2.1) | 5.0 (1.3) | 1.0 (1.0) |
| GeoIV | 9.3 (1.3) | 0.7 (0.9) | 1.0 (1.0) | 25.5 (0.0) | 1.5 (0.2) | 1.0 (1.4) |
| GeoV | 4.3 (1.1) | 0.2 (0.3) | 1.0 (1.0) | 10.0 (1.0) | 0.4 (0.0) | 1.0 (1.0) |