# Peer review of "Dynamic response of Antarctic Peninsula Ice Sheet to collapse of Larsen C and George VI ice shelves"

_The Cryosphere, 2018_

## Referee Comment (RC1) · Anonymous Referee #1 · 15 Mar 2018

**1   Summary statement**

This manuscript presents simulations from different ice sheet models showing the impact of a potential collapse of Larsen C and George VI ice shelves on the tributary glaciers feeding them. They investigate the case of a sudden and gradual collapse, and assess the impact of different model parameters (grid resolution, sliding law, ...) on the results. They show that changes in the Larsen C ice shelf have limited impact on its tributary glaciers, as this ice shelf provides a limited amount a buttressing. A collapse of George VI ice shelf on the other hand would have a much larger impact, as it provides more buttressing to its tributary glaciers and these glaciers are resting

on bedrock with retrograde slopes inland, making them prone to the marine ice sheet instability.

The results presented in this manuscript are novel and interesting, showing the very different response of glaciers in two basins, in terms of grounding line retreat and contribution to sea level change. It is great to see that this study is based on three different models, however one of them is exactly state of the art, and present results largely different to the other two, so it would be great to discuss this point and conclude on the possibility (or not) to use such simple models to investigate dynamic changes of Antarctic glaciers. Furthermore, there is not much discussion in this manuscript, just a description of the results, so it would be good to see a more substantial discussion added, including the impact of the different choices make in the model such as the sliding law used, the model resolution, and the agreement between models or between scenarios. The paper is well written and clear, except for the two tables and their captions, which are quite confusing. Below are some more detailed comments.

**2  Major comments**

I think it would be great to add "potential" in the title ("... to a potential collapse ..."), to highlight that this is just a possibility, or a future event. I think this is important given the recent collapse of the Larsen C iceberg, as it might confuse some people to talk about the collapse of Larsen C.

I found it confusing that the experiments are described one after the other as the text goes (new friction laws, different resolutions, ...). It would help to list all the experiments done in section 2 (maybe in 2.5 mention the additional experiments), or add a table with the list of experiments, so that the reader knows ahead of time what to expect.

In section 2.4, it is stated that the models should start with an initial state as close as possible from a steady-state. I disagree with this statement; the goal of the initial-

ization is to get as close as possible to the conditions at a given time, including the thinning rate observed at this time. Removing this thinning/thickening rate can lead to an underestimation/overestimation of the changes simulated, especially as this kind of signal would probably take decades to fade out. Also, how large is the flux correction applied to the models and how does it impact the simulations and the conclusions of this paper.

p.10 l.2: I have a different interpretation of the Pattyn et al. (2013) paper. If steady-state grounding line positions are well captured with an internal flux condition, the paper states that "the short-time transient behavior is then incorrect" (abstract of Pattyn et al. (2013)). So such models might be less dependent to grid resolution but it does not mean that they are accurate.

Fig.7 shows that for some basins and variables, there is a good agreement between the PSU3D and BISICLES models for the different scenarios, while in other cases, there is a bigger difference between the two models that between the different scenarios. This should be better discussed, especially to highlight the reasons of these differences as well as the different cases. Section 3 describes these results, but there should be some discussion summarizing these findings.

Overall, there is no real discussion, just a description of the results. A proper discussion should include the current limitations of the models and future possible improvements, the impact of the different models compared to other parameters, such as the sliding law employed, the scenario chosen, or the bedrock used, with references to previous studies.

**3 Specific comments**

p.1 l.1: "past several": be more precise

p.1 l.13: "northerly limit": it would be great to explain this limit in a few words

Fig.1: "meters above sea level" is a bit confusing as all elevations are negative, maybe simply saying "in meters" would be enough. Also mention that the colorbar is truncated at 0, and maybe add the highest elevation in this area. The black polygons are not clear and can be confused with the grounding line position, consider using a different color or thick lines.

p.2 l.7: mention that happens on downward sloping bedrock elevation inland (not just on all marine based sectors)

p.2 l.9: remove "state-of-the-art" as I am not sure that the BAS-APISM model can be considered to be a state-of-the-art model ("simulates ice flow by solving the simplest permissible force basal approximation" p.3 l.7)

p.2 l.10: same as the title: add that you are talking about a potential collapse

p.3 l.10: "SIA is not valid at the grounding line", the problem here is rather that SIA is not valid on floating ice shelves and fast flowing ice streams.

p.3 l.16: "in assumed" → "is assumed"

p.3 l.34: Add sentences in the three model descriptions about the grid resolution (and grid resolution at the grounding line) employed in these three models.

p.4 Eq.1: What basal conditions (friction) is used for the BAS-APISM model?

p.5 l.20: What is $R$ exactly and how does it relate to the temperature in a few words?

p.5 l.24-25: How is this done (in a sentence or two)? Some technical explanations are missing.

p.6 l.18: ALBMAP is quite old, why not use the new BEDMAP2 or Huss and Farinotti, (2014) data for all the models?

p.6 l.23: As mentioned above, do you really want the simulations to start from a steadystate? Or from the current thinning/thickening rate? Why not correct this by adding the rate of thickness change instead of assuming that it is 0? And by the way, I don't agree that "After initialization, the sheet-shelf models should be in equilibrium". The models should represent the actual ice sheet state at the time captured by the initialization, so if the ice sheets where thinning, the initialization should capture and reproduce this initial thinning.

p.6 l.28: Adding this flux correction is fine, but you should show how large it is, and how large it is compared to the actual surface mass balance. Also, how different are the results if you don't include it? What are the impacts on the simulations?

p.7 Fig.2: Why not show the BAS-APISM model here? I have a hard time understanding what the basal boundary condition of this model is. Also should be "Black lines denote ..."

p.8 l.3: What resolutions are used? The list of experiments with their characteristics should be better detailed in the text.

p.9 l.12: Over what period does this change happens?

p.9 l.14-19: The initial conditions (ice velocity, thickness, elevation, rigidity, ...) also have an impact on the evolution of the glacier, as well as the numerical parameters (grid resolution, ...).

p.9 l.14-19: What about the BAS-APISM model?

p.10 Fig.4: Should be: "Upper panels (a,b) show ...", same for "Lower panels ..."

p.10 l.2: As mentioned above, the Pattyn et al. (2013) paper says that "the short-time transient behavior is then incorrect" for grounding line evolution captured with internal flux conditions.

p.11 Fig.5: Should be: "Black lines denote ...". Same for caption in Fig.6.

p.15 l.3: "A consequence of this is ..." → "A consequence is ..."

[Figure]

p.15 l.23: "most of grounding-line retreat" → "most of the grounding-line retreat"

Tables 1 and 2: the tables and their captions are quite confusing. Especially as all numbers reflect different time periods, and some variables are not standard (e.g., dGt/dt for mass change rate). Also why not use the same order as Fig.6 (BAS-APISM left, ...).

Table 2: Fig.A7 shows larger grounding line retreat for many glaciers (GeoIII, GeoIV, ...) with the Coulomb friction law, which does not seemed to be reflected in this table. But as I just mentioned above, I am quite confused by this table. I would also expect this increased grounding line retreat to transfer in more mass change for the Coulomb case. It would be simpler to have both BISICLES cases next to each other.

p.18: As mentioned previously, there is not much discussion, just a description of the results.

p.19 l.6: "vulnerability of ice-shelf ..." → "vulnerability to ice-shelf .."

Fig.A1: Y-axis label should be "Temp. bias" not "Temp.". Caption should detail bias which two quantities.

Fig.A2: Same as Fig.A1

Fig. A6: Caption should be "Upper panels (a,b) show ...". Same for "Lower panels ..."

Fig. A7: Simulations with Coulomb friction show a larger retreat, which is not captured in Table 2.

**4  References**

Huss, M., and D. Farinotti, A high-resolution bedrock map for the Antarctic Peninsula, Cryosphere, 8(4), doi:10.5194/tc-8-1261-2014, 2014.

Pattyn, F., et al., Grounding-line migration in plan-view marine ice-sheet models: results of the ice2sea MISMIP3d intercomparison, J. Glaciol., 59 (215), doi: 10.3189/2013JoG12J129, 2013.

---

## Referee Comment (RC2) · L. Favier (Referee) · 12 Apr 2018

April 12, 2018

**General comments**

This paper from Clemens Schannwell and his colleagues investigates, in a timely manner, the response of the Antarctic Penisula glaciers to a collapse of the Larsen C and Georges VI ice shelves. They use three types of numerical models of different complexity applied to two main experiments, supplemented by secondary experiments. Experiment 1 starts without ice shelves, as if an ice shelf collapse had already occurred, Experiment 2 starts with the current geometry and use a calving law and potentially leads to ice shelf collapse, depending on future scenarios, which are either RCP4.5 or RCP8.5. The secondary experiments helps to quantify uncertainties, and are built upon Experiment 2 to which was added mild or strong sub-shelf melting, and a last experiment use another dataset for ice geometry (the one of Huss and Farinotti 2014) instead of the classical Bedmap2 dataset used elsewhere in the study. All those experiments are simulated from today to 2300.

I find this study very interesting and I think this can be published with minor revisions. The fact of applying different types of ice sheet models, having not only different physics for time evolutive simulations but also different approaches for building the initial spin-up (inversion + relaxation), to the same case of study is not easy but it

makes the results more robust. The paper is globally well written apart from some details on which you will have specific comments below. The description of the methods is quite clear, even though it lacks the definition of Glen's flow law from which the authors could introduce the enhancing factor, which should be defined clearly.

I first have a couple of minor concerns:

- The Spin-up for BISICLES at a 1000m resolution looks odd from Figure3, the results doesn't seem to converge with the resolution (dvdt at 0 very rapidly for 4000, 2000 and 500m resolutions, but quite different for 1000m). Did you, by any chance, accidently shift, say, the colors for 4000m and 1000m ??? If not, could you make a few comments on that in the paper.
- I was not always sure about the type of SMB that you applied, for the spin-up but also for the experiments. You introduce the Albmap SMB but in the inversion process, not in the spin-up neither in the description of transients. Make it more clear in the text.
- Could you change the time origin in the evolutive plots to be 2000 instead of 0 ?
- Could you mention that sub-shelf melting is always 0 (if I understood correctly), apart from your two additional experiments
- I would be glad if you could indicate the position of the calving front in your maps of Experiment2 results, for 2100 and 2300 for instance, or maybe, if it's more easy, indicate the year of collapse in Figure7. This would strongly help the understanding of ice dynamics differences between Exp1 and Exp2.
- I would be in favor of adding a table to summarise all the experiments, including the main exp1 and Exp2 but also the three others.
- I would also be in favor of adding those results to Table2, and put the two tables in the supplementary, to help the reading and understanding of what has been done.
- Some assertions are not always correct in the reading of the results (see below)

The rest of my review is a series of specific comments and recommmendations, which

I would like to be followed as well.

**Specific comments**

Page1
l17: Could you mention those other mechanisms ?
l18: Could you add a word like "slightly" just before increased ? This is at least what I understand from the Jansen et al., 2015 study
Page2
Figure 1:
- The Y label should be "elevation [m a.s.l]"
- In the caption, "localities mentioned *in the text*"
- In the caption, remove "below sea level"
- In the caption, replace "Black polygons..." by something like "The grounded part of the ice sheet only is represented"
l8: "a tendency...": Here the way this instability works doesn't appear clearly. Nowhere there is written that you have deepening of the bedrock towards the interior, which is a necessary condition (at the necessary condition that bedrock is below sea level) to a MISI. Be more precise please.
Page3
l10: Why is the SIA not valid at the grounding line ?  could you mention the reason or/and add a citation here ?
l21: The way this is said, it is not clear whether the condition is imposed at the grounding line, to me at least... Could you rephrase.
Page4
l11: Is there a reason to take 0.5 specifically ? Tsai et al., advises f<0.6, Brondex et al., takes 0.5. Could you make a few comments on that.

Page5:

l9: So the criterion in Bisicles is Height surface crevasse + Height basal crevasse = ice surface ? this is not clear to me

l12: Do you need a capital H in historical ?

l29: You should have defined this enhancement factor before reaching this part. It is worth to detail the Glen's flow law equation somewhere above, from which you can easily define the enhancement factor.

Page6

l7: I am glad you showed Figure A3 (Hum, this figure reminds me Belgium...)

l11: m=1 ? Something is wrong here, or is this a typo ?

l18: You used SMB for the spin-up only (and alos the transient for , could you be more accurate here

Page7

Figure 2: Is that the PSU3D grid that we can distinguish in Figure2 ? Why is that so ?

l4: " In the first simulation (hereafter Experiment 1)" -> in Experiment 1 ? l5: This is not clear what you took for SMB in Experiment 1 ?

Page8

Figure3:

- Isn't that curious that the 1000m resolution results for BISICLES are outlying compared to the others ? I would have expected the results to converge when getting closer to 500m resolution, but it is not the case here. Could you discuss that ?

- The solution of PSU3D is not really converging. The model can't be applied at a lower resolution ? Could you add few comments on that if you find it relevant. l1: "best available": I don't think this is relevant to say so, no offense...

Page9

l5: Results and discussion

l12: Sea-level rise by 2100 ?

l17: The differences in terms of sea level response may be due to those large differences that you have between friction coefficients fields ?

l18: Could you maybe detail the relevant specific differences ?

Page10

Figure4: Could you start your time scale at 2000 ?

l1: where do we see the dependence to grid resolution in Figure A4 ? This is rather observed in Figure A6. And according to Figure A6, there is also a sea level contribution dependency to grid resolution in PSU3D. Moreover, this is true that this dependency is small for Georges VI (not absent though) and comparable to BISICLES for Larsen C. Could you rephrase here.

l4: you definitely refer to Figure A6

Page11

Figure5: It seems that the 2000 grounding line is slightly different between PSU3D and BISICLES. The differences are difficult to quantify, so could you maybe write down somewhere the maximum difference between initial grounding lines for the two models?

l10: A necessary condition to have a MISI is a retrograde bed slope, insofar as you have a marine based basin as well. You thus need to replace "mostly marine-based outlet" by "retrograde bed slope something..."

l13: Could you discuss this a bit more. There is this paper from Gudmundsson et al., in 2012 and Gudmundsson in 2013 about the buttressing provided by an ice shelf to its upstream glacier as a function of the grounding line gate width...

Page12

l3: "dependent"

l7: Remove "fixed calving front simulation, immediate shelf-collapse scenario" and keep "Experiment 1"

l10: Remove "just"

l10: "the larger" -> "the largest"

Page13

Figure7: Time between 2000 and 2300

l1: "projections ... agree well": this is not really what I see in Figure7a, where the

two models may agree for the first 50 to 70 years, but not really for the rest of the simulations. Could you rephrase ?

l4: "BISICLES projects no slr for RCP4.5": more correct would be to say "small" or "limited" instead of "no", especially for Larsen C

Page14

l1: "Both sheet-shelf models project similar sea-level rise by the mid 22nd century in Experiment 2": Again, I don't agree with this sentence, after 250 years Larsen C ice loss is 0.25 and 0.6 for BISICLES4.5 and PSU3D4.5, and this is not the only example where I find differences where you write the opposite (see above). You should rewrite here.

l2: What do you mean by "forced back" ? Does it simply mean that the grounding line retreat ? Could you rephrase ?

l3: "because a fixed calving front": Here I dont 'understand, the fact of having a fixed calving front does not prevent the retreat of the grounding line. You need to rephrase.

l1 to l14: For this paragraph, this is not cristal clear to me if you talk about Georges VI or Larsen C ice shelf. Can you make the text more clear please.

l25: Don't understand this sentence. you say that your model show a strong dependence to what ? Calving criteria or sub-shelf melting ?

Page15

l3: Ok, this is another experiment. I recommend to add a table to summarize all the experiments

l8: five more drainage basins, means not the LarI to LarV and GeoI to GeoV ? could you indicate in a figure which are the supplementary basins that you accounted for ? Maybe put it into Figure8 ?

Page17

Table1: What did you put in parenthesis from the dGL and dGt/dt columns ? Does it correspond to the year you had the maximum speed up ? You need to write it down then.

Table1 and Table2: Would you like to move those 2 tables in the Supplementary ? I

have the feeling that it affects the reading of the paper...
Page19
l4: You definitely need to show your results with the Fuss and Farinotti geometry

Suplementary
Figure A3: Could you explain why you chose $\lambda_C = 10^{-1}$ ? It doesn't seem that obvious looking at A3c. Stupid question maybe, why is there a jump between $10^{-1}$ and $10^1$ (I mean no $10^0$ appearing) ?
FigureA6: Time origin should be 2000

[Figure]

---

## Author Comment (AC1) · 9 May 2018

**We thank both referees for their thoughtful and thorough reviews of our paper. We appreciate you taking the time to complete these reviews and welcome your helpful comments. We have revised the manuscript to address your review comments (see below). Throughout this response to review document your (referee review) comments are provided in regular, non- italic font text, our response comments are provided in red font (as here).**

**Reviewer 1:**

**1 Summary statement**

This manuscript presents simulations from different ice sheet models showing the im- pact of a potential collapse of Larsen C and George VI ice shelves on the tributary glaciers feeding them. They investigate the case of a sudden and gradual collapse, and assess the impact of different model parameters (grid resolution, sliding law, ...) on the results. They show that changes in the Larsen C ice shelf have limited impact on its tributary glaciers, as this ice shelf provides a limited amount a buttressing. A collapse of George VI ice shelf on the other hand would have a much larger impact, as it provides more buttressing to its tributary glaciers and these glaciers are resting on bedrock with retrograde slopes inland, making them prone to the marine ice sheet instability.

The results presented in this manuscript are novel and interesting, showing the very different response of glaciers in two basins, in terms of grounding line retreat and contribution to sea level change. It is great to see that this study is based on three different models, however one of them is exactly state of the art, and present results largely different to the other two, so it would be great to discuss this point and conclude on the possibility (or not) to use such simple models to investigate dynamic changes of Antarctic glaciers. Furthermore, there is not much discussion in this manuscript, just a description of the results, so it would be good to see a more substantial discussion added, including the impact of the different choices make in the model such as the sliding law used, the model resolution, and the agreement between models or between scenarios. The paper is well written and clear, except for the two tables and their captions, which are quite confusing. Below are some more detailed comments.

**2 Major comments**

I think it would be great to add "potential" in the title (”... to a potential collapse ...”), to highlight that this is just a possibility, or a future event. I think this is important given the recent collapse of the Larsen C iceberg, as it might confuse some people to talk about the collapse of Larsen C.

**We agree with the reviewer and changed the title accordingly.**

I found it confusing that the experiments are described one after the other as the text goes (new friction laws, different resolutions, ...). It would help to list all the experiments done in section 2 (maybe in 2.5 mention the additional experiments), or add a table with the list of experiments, so that the reader knows ahead of time what to expect.

**We agree and have added a table listing all perturbation experiments including sensitivity simulations as well as grid resolution to the section 'Experimental Design' (Section 2.5).**

In section 2.4, it is stated that the models should start with an initial state as close as possible from a steady-state. I disagree with this statement; the goal of the initialization is to get as close as possible to the conditions at a given time, including the thinning rate observed at this time. Removing this thinning/thickening rate can lead to an

underestimation/overestimation of the changes simulated, especially as this kind of signal would probably take decades to fade out. Also, how large is the flux correction applied to the models and how does it impact the simulations and the conclusions of this paper.

**The reviewer is correct that if the ice sheet is not in steady state then there should be a thinning/thickening rate after the initialisation. As the goal of this study is to tease out the contribution from ice-shelf removal to sea-level rise projections, we want an ice-sheet geometry that does not change over time. This is why we apply the synthetic mass balance to keep the geometry as close as possible to the initial geometry. To make this clearer in the manuscript we changed the first paragraph of the 'Spin-Up' section. It reads as follows: "Following initialisation the sheet-shelf models should be close to equilibrium if the ice sheet is close to steady state, providing dh/dt = 0. However, owing to data inconsistencies and in part a violation of this steady-state assumption this condition is not fulfilled, requiring a spin-up or relaxation simulation to reach a steady state for each model. To tease out the sea-level rise contributions from ice-shelf removal and facilitate comparison across all three ice-sheet models, the employed spin-up approach aims to keep the ice sheet geometry as close as possible to the initial geometry. "**

p.10 l.2: I have a different interpretation of the Pattyn et al. (2013) paper. If steady-state grounding line positions are well captured with an internal flux condition, the paper states that "the short-time transient behavior is then incorrect" (abstract of Pattyn et al. (2013)). So such models might be less dependent to grid resolution but it does not mean that they are accurate.

**We agree with the reviewer that the short-term transient behaviour of these hybrid models with an internal flux boundary condition may not be correct. However, in most of our perturbation simulations quasi steady-state is reached with PSU3D, meaning that steady-state grounding line positions should agree better between PSU3D and BISICLES after 300 years. The sentence is just stating that grid dependence is much reduced in PSU3D in comparison to BISICLES. As the Pattyn et al. (2013) paper is not the best citation for this, we have removed it from the revised manuscript.**

Fig.7 shows that for some basins and variables, there is a good agreement between the PSU3D and BISICLES models for the different scenarios, while in other cases, there is a bigger difference between the two models that between the different scenarios. This should be better discussed, especially to highlight the reasons of these differences as well as the different cases. Section 3 describes these results, but there should be some discussion summarizing these findings.

**We have added a paragraph to the discussion section to discuss the differences in results more in depth. It reads: "We attribute the good agreement across both models for Larsen C to the fact that the area of the marine-based sectors is limited in this domain (2.1 mm contained in marine-based sectors) due to the very mountainous bedrock topography constraining potential grounding-line retreat. This is supported by all simulations across all ice-sheet models as even under a wide range of different forcings the Larsen C embayment does not contribute more than 4.2 mm by 2300. The greater potential to initiate grounding-line retreat is presented by George VI Ice Shelf where much of the ice sheet is marine based with retrograde sloping bedrock topography (Figure 1b). As this large grounding-line retreat is only initiated in the BISICLES simulation, large differences in sea-level rise projections occur. The most likely scenario for this differing behaviour is due to the difference in the inferred basal traction coefficient fields that affects each model's response to ice-shelf removal. PSU3D predicts much stickier bedrock conditions in the George VI embayment than BISICLES (Figure 2). These sticky bedrock conditions result in**

**little acceleration of the major outlet glaciers following ice-shelf breakup. This in turn means that the calving law applied to only floating ice cells cannot drive the initial retreat into the marine based sectors as the outlet glaciers do not thin sufficiently to form a floating ice tongue. In contrast in the RCP8.5 BISICLES simulation for George VI, speed-up in response to ice-shelf breakup leads to enhanced dynamic thinning of the main outlet glaciers. This thinning in conjunction with the calving law drives the calving front into the marine-based sectors where further retreat is initiated by a combination of the marine ice-sheet instability and the meltwater driven calving law, resulting in the simulated much higher sea-level rise projections."**

Overall, there is no real discussion, just a description of the results. A proper discussion should include the current limitations of the models and future possible improvements, the impact of the different models compared to other parameters, such as the sliding law employed, the scenario chosen, or the bedrock used, with references to previous studies.

**We have added a paragraph to section 3.3. that discusses and highlights the model limitation, key parameter uncertainties and improvements for future studies. It reads: "In addition, our experiments show that for simulations of grounding-line motion in response to ice-shelf breakup sheet-shelf models are necessary. The simple model BAS-APISM fails to reproduce the results of the sheet-shelf models due to the simplified physics. Even across sheet-shelf models differences in model physics, model initialisation, calving law implementation and other numerics (e.g. meshing) can lead to substantially different projections under the same forcing (Figure A5). Sea-level rise projections are most sensitive to the choice of sliding law and bedrock geometry. The peninsula is not the only region where these parameters highly affect decadal to centennial sea-level rise projections as similar conclusions were drawn from modelling of outlet glaciers in the Amundsen Sea embayment (Nias et al., 2018). The wide range of sea-level rise responses to different forcing parameters underlines the need for perturbed ensembles to explore key parameter uncertainties (e.g. basal sliding law) for sea-level rise projections in greater detail for the peninsula region. Owing to the increase in computer power these type of ensemble projections have become feasible at the regional (e.g. Nias et al., 2016) and continental scale (e.g. DeConto and Pollard, 2016).**

**3 Specific comments**

p.1 l.1: "past several": be more precise
**changed to "past five decades"**

p.1 l.13: "northerly limit": it would be great to explain this limit in a few words
**we added "… determined by the -9°C mean annual isotherm…"**

Fig.1: "meters above sea level" is a bit confusing as all elevations are negative, maybe simply saying "in meters" would be enough. Also mention that the colorbar is truncated at 0, and maybe add the highest elevation in this area. The black polygons are not clear and can be confused with the grounding line position, consider using a different color or thick lines.
**We have changed the text and figure accordingly. Black polygon lines have been made thicker and we simply state now "…elevations below sea level in meters".**

p.2 l.7: mention that happens on downward sloping bedrock elevation inland (not just on all marine based sectors)
**We added "…and retrograde sloping bedrock topography…"**

p.2 l.9: remove "state-of-the-art" as I am not sure that the BAS-APISM model can be considered to be a state-of-the-art model ("simulates ice flow by solving the simplest permissible force basal approximation" p.3 l.7)

**Removed**

p.2 l.10: same as the title: add that you are talking about a potential collapse

**Done**

p.3 l.10: "SIA is not valid at the grounding line", the problem here is rather that SIA is not valid on floating ice shelves and fast flowing ice streams.

**Changed to: "As the SIA is not valid for floating ice shelves, …"**

p.3 l.16: "in assumed" → "is assumed"

**Changed**

p.3 l.34: Add sentences in the three model descriptions about the grid resolution (and grid resolution at the grounding line) employed in these three models.

**We have added this information to the table where all simulations are listed.**

p.4 Eq.1: What basal conditions (friction) is used for the BAS-APISM model?

**We added a sentence specifying that due to the linearisation there is no need to specify whether or not basal sliding is occurring. It reads: "Due to the linearisation of the evolution equations in BAS-APISM, there is no need to specify whether or not basal sliding is occurring. All rates are determined by the ice flux which is directly derived from the data."**

p.5 l.20: What is R exactly and how does it relate to the temperature in a few words?

**We added: "This formula scales surface melt exponentially with mean DJF near surface temperatures …"**

p.5 l.24-25: How is this done (in a sentence or two)? Some technical explanations are missing.

**We have added a sentence saying: "This is accomplished through the computation of balance fluxes."**

p.6 l.18: ALBMAP is quite old, why not use the new BEDMAP2 or Huss and Farinotti, (2014) data for all the models?

**The difference in ice volume and bedrock topography between ALBMAP and BEDMAP2 for the Antarctic Peninsula is rather small (<15%) and the Huss and Farinotti (2014) dataset is only available for the Larsen C domain as it does not cover the southern part of the peninsula. To gauge the importance of differences in bedrock topography, we carried out the sensitivity simulation with BISICLES for the Larsen C domain. As the difference between BEDMAP2 and the Huss and Farinotti (2014) dataset is large (~100% in ice volume below sea level), large differences propagate into the magnitude of the sea-level rise projections.**

p.6 l.23: As mentioned above, do you really want the simulations to start from a steady-state? Or from the current thinning/thickening rate? Why not correct this by adding the rate of thickness change instead of assuming that it is 0? And by the way, I don't agree that "After initialization, the sheet-shelf models should be in equilibrium". The models should represent the actual ice sheet state at the time captured by the initialization, so if the ice sheets where thinning, the initialization should capture and reproduce this initial thinning.

p.6 l.28: Adding this flux correction is fine, but you should show how large it is, and how large it is compared to the actual surface mass balance. Also, how different are the results if you don't include it? What are the impacts on the simulations?

**See reply above. In brief, in order to really tease out the contributions that come from ice-shelf removal alone, we desire all other signals to be as close as possible to zero. This is**

**why we apply the synthetic mass balance to keep the ice-sheet as close as possible to its initial geometry. Of course the results of our simulations would be different without the additional synthetic surface mass balance forcing, but comparing our projections with and without this flux term is not the goal of this study.**

p.7 Fig.2: Why not show the BAS-APISM model here? I have a hard time understanding what the basal boundary condition of this model is. Also should be "Black lines denote ..."

**Due to the linearization of the evolution equation there is no need to specify whether or not basal sliding is occurring and balance fluxes are used to initialise the model.**

p.8 l.3: What resolutions are used? The list of experiments with their characteristics should be better detailed in the text.

**We have added a table with all simulations and their respective resolution to section 2.5.**

p.9 l.12: Over what period does this change happens?

**We added: "… averaged over 300 years"**

p.9 l.14-19: The initial conditions (ice velocity, thickness, elevation, rigidity, ...) also have an impact on the evolution of the glacier, as well as the numerical parameters (grid resolution, ...).

**We added: "This discrepancy between the sheet-shelf models may be attributed to a combination of differences in initialisation and that PSU3D is not as close to steady-state as BISICLES following initialisation and spin-up."**

p.9 l.14-19: What about the BAS-APISM model?

**As there is no time-dependent grounding-line migration after the initial perturbation, we focus on the sheet shelf models, but state at the end of the paragraph that BAS-APISM projects similar magnitudes of sea-level rise, but the spatial thinning pattern is very different to the sheet-shelf models.**

p.10 Fig.4: Should be: "Upper panels (a,b) show ...", same for "Lower panels ..."

**Changed**

p.10 l.2: As mentioned above, the Pattyn et al. (2013) paper says that "the short-time transient behavior is then incorrect" for grounding line evolution captured with internal flux conditions.

**See reply above. We have removed the Pattyn et al. (2013) citation.**

p.11 Fig.5: Should be: "Black lines denote ...". Same for caption in Fig.6.

**Changed**

p.15 l.3: "A consequence of this is ..." → "A consequence is ..."

**Changed**

p.15 l.23: "most of grounding-line retreat" → "most of the grounding-line retreat"

**Changed**

Tables 1 and 2: the tables and their captions are quite confusing. Especially as all numbers reflect different time periods, and some variables are not standard (e.g., dGt/dt for mass change rate). Also why not use the same order as Fig.6 (BAS-APISM left, ...).

**As suggested by the second reviewer, we have moved the tables to the Supplementary material as they disrupted the flow of the paper. We also changed the dGt/dt to the correct dM/dt and the respective units (if not unitless) are provided in the table caption. We also changed the order of the columns to follow the order of Figures 6 and 8 for Experiment 1 and Experiment 2, respectively. Moreover, we now present a separate table for George VI and Larsen C basins and added the Coulomb sliding BISICLES simulation for Experiment 1 to Table S1.**

Table 2: Fig.A7 shows larger grounding line retreat for many glaciers (GeoIII, GeoIV, ...) with the Coulomb friction law, which does not seemed to be reflected in this table. But as I just

mentioned above, I am quite confused by this table. I would also expect this increased grounding line retreat to transfer in more mass change for the Coulomb case. It would be simpler to have both BISICLES cases next to each other.

**Due to the confusing layout, the reviewer confused Experiments 1 and 2 here. They do show larger grounding-line retreat but only in comparison to other simulations from Experiment 1, but not in comparison to Experiment 2. We added the Coulomb sliding BISICLES simulation for Experiment 1 to Table S1 to make this clearer.**

p.18: As mentioned previously, there is not much discussion, just a description of the results.
**See reply above.**

p.19 l.6: "vulnerability of ice-shelf ..." → "vulnerability to ice-shelf .."
**Changed**

Fig.A1: Y-axis label should be "Temp. bias" not "Temp.". Caption should detail bias which two quantities.
Fig.A2: Same as Fig.A1

**Changed and added "…in relation to ERA-Interim."**

Fig. A6: Caption should be "Upper panels (a,b) show ...". Same for "Lower panels ..."
**Changed**

Fig. A7: Simulations with Coulomb friction show a larger retreat, which is not captured in Table 2.
**It is captured. See reply above.**

**Reviewer 2:**
General comments
April 12, 2018
This paper from Clemens Schannwell and his colleagues investigates, in a timely manner, the response of the Antarctic Penisula glaciers to a collapse of the Larsen C and Georges VI ice shelves. They use three types of numerical models of different complexity applied to two main experiments, supplemented by secondary experiments. Experiment 1 starts without ice shelves, as if an ice shelf collapse had already occurred, Experiment 2 starts with the current geometry and use a calving law and potentially leads to ice shelf collapse, depending on future scenarios, which are either RCP4.5 or RCP8.5. The secondary experiments helps to quantify uncertainties, and are built upon Experiment 2 to which was added mild or strong sub-shelf melting, and a last experiment use another dataset for ice geometry (the one of Huss and Farinotti 2014) instead of the classical Bedmap2 dataset used elsewhere in the study. All those experiments are simulated from today to 2300.
I find this study very interesting and I think this can be published with minor revisions. The fact of applying different types of ice sheet models, having not only different physics for time evolutive simulations but also different approaches for building the initial spin-up (inversion + relaxation), to the same case of study is not easy but it makes the results more robust. The paper is globally well written apart from some details on which you will have specific comments below. The description of the methods is quite clear, even though it lacks the definition of Glen's flow law from which the authors could introduce the enhancing factor, which should be defined clearly.
I first have a couple of minor concerns:
- The Spin-up for BISICLES at a 1000m resolution looks odd from Figure3, the results doesn't seem to converge with the resolution (dvdt at 0 very rapidly for 4000, 2000 and 500m resolutions, but quite different for 1000m). Did you, by any chance, accidently shift, say, the

colors for 4000m and 1000m??? If not, could you make a few comments on that in the paper.

**We did not expect convergence in the spin-up simulations by increasing the mesh resolution. These simulations all use a synthetic surface mass balance that is derived from their respective model grids after one timestep. In some of the simulations, the model drift and the remaining thinning/thickening signal following initialisation may not be as well captured after one timestep as in others (e.g. 2000 m vs. 1000 m BISICLES). In addition, the initialisations were performed also on different grids (1 km for BISICLES and 5 km for PSU3D). All in all, these factors most likely lead to the slightly different dV/dt patterns that are shown in Figure 3. Considering that steady-states with real world geometries are difficult to achieve, we are satisfied with how close they are to steady state after relaxation. PUS3D could not be run at higher resolution as it became unstable at resolutions of < 1000 m. Therefore, we chose 1000 m.**

- I was not always sure about the type of SMB that you applied, for the spin-up but also for the experiments. You introduce the Albmap SMB but in the inversion process, not in the spin-up neither in the description of transients. Make it more clear in the text.

**We have added a sentence to section 2.4 that clarifies that we use the synthetic mass balance for all experiments. It reads: "This synthetic mass balance is applied in all spin-up and perturbation simulations."**

- Could you change the time origin in the evolutive plots to be 2000 instead of 0?

**Changed accordingly**

- Could you mention that sub-shelf melting is always 0 (if I understood correctly), apart from your two additional experiments.

**We added a sentence to section 2.5. to clarify this. It reads: "Moreover, ocean melting is set to zero in the perturbation experiments unless stated otherwise."**

- I would be glad if you could indicate the position of the calving front in your maps of Experiment2 results, for 2100 and 2300 for instance, or maybe, if it's more easy, indicate the year of collapse in Figure7. This would strongly help the understanding of ice dynamics differences between Exp1 and Exp2.

**We think that adding the calving front to Figure 8 at these two timesteps would clutter up the Figure too much. Moreover, the calving front in 2300 is very close to the grounding-line that is drawn in Figure 8 anyway. Rather than visualise the calving front with lines in Figure 8, we find the ice-shelf area loss plots (Figure 7e, f) more informative as the evolution of ice-shelf retreat can be tracked better over time. In addition to these plots, we added a sentence that states that once the ice shelf has collapsed, grounding-line and calving front are almost in identical locations.**

- I would be in favor of adding a table to summarise all the experiments, including the main exp1 and Exp2 but also the three others.

**We have added a table listing all experiment and the resolutions at which they were run to section 2.5.**

- I would also be in favor of adding those results to Table2, and put the two tables in the supplementary, to help the reading and understanding of what has been done.

**Yes, we moved the tables to the Supplementary as they disrupted the flow of the paper. We also changed the dGt/dt to the correct dM/dt and the respective units (if not unitless) are provided in the table caption. We also changed the order of the columns to follow the order of Figures 6 and 8 for Experiment 1 and Experiment 2 respectively. Moreover, we now present a separate table for George VI and Larsen C basins and added the Coulomb sliding BISICLES simulation for Experiment 1 to Table S1.**

- Some assertions are not always correct in the reading of the results (see below)

The rest of my review is a series of specific comments and recommmendations, which would like to be followed as well.

**Specific comments**

Page1

l17: Could you mention those other mechanisms?

**We added "… such as ice-shelf thinning, fracturing, and weakening of shear margins …"**

l18: Could you add a word like "slightly" just before increased? This is at least what I understand from the Jansen et al., 2015 study

**Done**

Page2

Figure 1:

- The Y label should be "elevation [m a.s.l]"
- In the caption, "localities mentioned in the text"
- In the caption, remove "below sea level"
- In the caption, replace "Black polygons..." by something like "The grounded part of the ice sheet only is represented"

**We have changed to "localities in the text" and followed the suggestions from reviewer 1. Black polygon lines have been made thicker and we simply state now "…elevations below sea level in meters".**

l8: "a tendency...": Here the way this instability works doesn't appear clearly. Nowhere there is written that you have deepening of the bedrock towards the interior, which is a necessary condition (at the necessary condition that bedrock is below sea level) to a MISI. Be more precise please.

**We added "…and retrograde sloping bedrock topography…"**

Page3

l10: Why is the SIA not valid at the grounding line? could you mention the reason or/and add a citation here ?

**We have added a citation to (Hutter, 1983)**

l21: The way this is said, it is not clear whether the condition is imposed at the grounding line, to me at least... Could you rephrase.

**We have added "… is employed at the grounding line."**

Page4

l11: Is there a reason to take 0.5 specifically? Tsai et al., advises f<0.6, Brondex et al., takes 0.5. Could you make a few comments on that.

**Most ice-sheet modelling studies have used a value of f=0.5 (see Asay-Davis et al. 2016, Brondex et al. 2017, Nias et al. 2018). To keep in line with this, we chose the same value for our simulations.**

Page5:

l9: So the criterion in Bisicles is Height surface crevasse + Height basal crevasse = ice surface? this is not clear to me

**We have rewritten to read: "…reaches the distance from ice surface to the waterline."**

l12: Do you need a capital H in historical?

**Changed to lower case.**

l29: You should have defined this enhancement factor before reaching this part. It is worth to detail the Glen's flow law equation somewhere above, from which you can easily define the enhancement factor.

**Added to the section 2.1.**

Page6

l7: I am glad you showed Figure A3 (Hum, this figure reminds me Belgium...)

**We added a sentence to acknowledge this. It reads: "The layout was inspired by Berger et al. (2016)."**

l11: m=1 ? Something is wrong here, or is this a typo?

**We added "… for the inversion simulation." We then recompute the basal drag coefficient field from the inversion (linear sliding) for the other sliding laws used in the perturbation experiments.**

l18: You used SMB for the spin-up only (and alos the transient for, could you be more accurate here

**This was ambiguous. We removed surface mass balance here and added a sentence to section 2.4 that clarifies that we use the synthetic mass balance for all experiments. It reads: "This synthetic mass balance is applied in all spin-up and perturbation simulations."**

Page7

Figure 2: Is that the PSU3D grid that we can distinguish in Figure2? Why is that so?

**We do not know what the reviewer means by this. No grids are shown in Figure 2. Only the outline of our model domain (black polygon) is visible.**

l4: " In the first simulation (hereafter Experiment 1)" -> in Experiment 1?

**Changed.**

l5: This is not clear what you took for SMB in Experiment 1?

**We added a sentence to section 2.4 that clarifies that we use the synthetic mass balance for all experiments. It reads: "This synthetic mass balance is applied in all spin-up and perturbation simulations."**

Page8

Figure3:

- Isn't that curious that the 1000m resolution results for BISICLES are outlying compared to the others? I would have expected the results to converge when getting closer to 500m resolution, but it is not the case here. Could you discuss that?

- The solution of PSU3D is not really converging. The model can't be applied at a lower resolution? Could you add few comments on that if you find it relevant.

**We did not expect convergence in the spin-up simulations by increasing the mesh resolution. These simulations all use a synthetic surface mass balance that is derived from their respective model grids after one timestep. In some of the simulations, the model drift and the remaining thinning/thickening signal following initialisation may not be as well captured after one timestep as in others (e.g. 2000 m vs. 1000 m BISICLES). In addition, the initialisations were performed also on different grids (1 km for BISICLES and 5 km for PSU3D). All in all, these factors most likely lead to the slightly different dV/dt patterns that are shown in Figure 3. Considering that steady-states with real world geometries are difficult to achieve, we are satisfied with how close they are to steady state after relaxation. PUS3D could not be run at higher resolution as it became unstable at resolutions of < 1000 m. Therefore, we chose 1000 m.**

l1: "best available": I don't think this is relevant to say so, no offense...

**Deleted.**

Page9

l5: Results and discussion

l12: Sea-level rise by 2100?

**For clarity we added: "…by 2300."**

l17: The differences in terms of sea level response may be due to those large differences that you have between friction coefficients fields?

**We have rewritten this sentence. It reads: "This discrepancy between the sheet-shelf models may be attributed to a combination of differences in initialisation, inferred basal traction fields, and that PSU3D is not as close to steady-state as BISICLES following initialisation and spin-up."**

l18: Could you maybe detail the relevant specific differences?

**We have extended this paragraph to accommodate this. It reads: "Such a response has been previously attributed to differences in the underlying model physics (L1L2, A-HySSA). Using synthetic geometries, A-HySSA models have shown to be more sensitive to grounding-line advance as well as retreat. These differences are most likely caused by the neglecting of vertical shearing terms in the pure membrane ice-sheet models (Pattyn et al., 2013)."**

Page10

Figure4: Could you start your time scale at 2000?

**Changed.**

l1: where do we see the dependence to grid resolution in Figure A4? This is rather observed in Figure A6.

**Changed.**

And according to Figure A6, there is also a sea level contribution dependency to grid resolution in PSU3D. Moreover, this is true that this dependency is small for Georges VI (not absent though) and comparable to BISICLES for Larsen C. Could you rephrase here.

**We rephrased to: "… much reduced …"**

l4: you definitely refer to Figure A6

**Yes, we changed it accordingly.**

Page11

Figure5: It seems that the 2000 grounding line is slightly different between PSU3D and BISICLES. The differences are difficult to quantify, so could you maybe write down somewhere the maximum difference between initial grounding lines for the two models?

**We state in section 3.2 that the effect of the more advanced grounding-line position in PSU3D for the Larsen C domain accounts for a sea-level equivalent of 0.28 mm. We find this number more informative than a maximum difference in grounding-line position and therefore would like to keep it.**

l10: A necessary condition to have a MISI is a retrograde bed slope, insofar as you have a marine based basin as well. You thus need to replace "mostly marine-based outlet" by "retrograde bed slope something..."

**Changed**

l13: Could you discuss this a bit more. There is this paper from Gudmundsson et al., in 2012 and Gudmundsson in 2013 about the buttressing provided by an ice shelf to its upstream glacier as a function of the grounding line gate width...

**This is a good point. We have extended the paragraph to discuss this now. It reads: "These findings suggest that stabilising forces such as basal and lateral drag may provide enough resistance for the ice sheet in western Palmer Land to remain in a stable configuration following the initial response to ice-shelf collapse. This is supported by earlier modelling studies with idealised geometries, showing that the magnitude of grounding-line retreat is a function of the retrograde sloping channel width (Gudmundsson et al., 2012, Gudmundsson et al., 2013). The smaller the channel width, the less retreat was simulated (Gudmundsson et al., 2012). Considering the small size of the drainage basins in the**

**peninsula region with channel widths <30 km, the remaining lateral buttressing from shear margins likely impedes any runaway grounding-line retreat."**

Page12

l3: "dependent"

**Fixed.**

l7: Remove "fixed calving front simulation, immediate shelf-collapse scenario" and keep "Experiment 1"

**Changed.**

l10: Remove "just"

**Removed.**

l10: "the larger" -> "the largest"

**We think "the larger" is correct.**

Page13

Figure7: Time between 2000 and 2300

**Changed**

l1: "projections ... agree well": this is not really what I see in Figure7a, where the two models may agree for the first 50 to 70 years, but not really for the rest of the simulations. Could you rephrase?

**We added "…reasonably well …"**

l4: "BISICLES projects no slr for RCP4.5": more correct would be to say "small" or "limited" instead of "no", especially for Larsen C

**Changed to "little"**

Page14

l1: "Both sheet-shelf models project similar sea-level rise by the mid 22nd century in Experiment 2": Again, I don't agree with this sentence, after 250 years Larsen C ice loss is 0.25 and 0.6 for BISICLES4.5 and PSU3D4.5, and this is not the only example where I find differences where you write the opposite (see above). You should rewrite here.

**Sorry this statement was meant for the George VI embayment. We have added a qualifier at the end of the sentence: "… for the George VI domain…" We also meant 2150 with "the mid 22nd century". To avoid any confusion, we changed this phrase to 2150.**

l2: What do you mean by "forced back"? Does it simply mean that the grounding line retreat? Could you rephrase?

**Yes. We changed this to read: "…grounding lines retreat further back into …"**

l3: "because a fixed calving front": Here I dont 'understand, the fact of having a fixed calving front does not prevent the retreat of the grounding line. You need to rephrase.

**Yes, we have rephrased it. After ice-shelf collapse grounding line and calving front are in almost identical locations. To highlight this, we added: "After ice-shelf collapse, grounding line and calving front for all drainage basins are almost in identical locations."**

l1 to l14: For this paragraph, this is not cristal clear to me if you talk about Georges VI or Larsen C ice shelf. Can you make the text more clear please.

**Yes, it was not very clear. We rewrote sections of the paragraph and also added a paragraph to discuss the difference in projections across the models.**

l25: Don't understand this sentence. you say that your model show a strong dependence to what? Calving criteria or sub-shelf melting

**We rephrased this to read: "…in other words, it is ice-shelf break-up in combination with the calving criteria that dominates our results."**

Page15

l3: Ok, this is another experiment. I recommend to add a table to summarize all the experiments

**Done. See above.**

l8: five more drainage basins, means not the LarI to LarV and GeoI to GeoV? could you indicate in a figure which are the supplementary basins that you accounted for ? Maybe put it into Figure8?

**Apologies this was ambiguous. We indeed mean the basins LarI to LarV and GeoI to GeoV. This sentence was changed to improve clarity. It reads: "To further assess the impact of ice-shelf break-up, five drainage basins from the Larsen C embayment (LarI-LarV, Figure 8) and George VI embayment (GeoI-GeoV, Figure 8) were selected for additional analysis."**

Page17

Table1: What did you put in parenthesis from the dGL and dGt/dt columns? Does it correspond to the year you had the maximum speed up? You need to write it down then.

Table1 and Table2: Would you like to move those 2 tables in the Supplementary? I have the feeling that it affects the reading of the paper...

**Yes, we moved the tables to the Supplementary as they disrupted the flow of the paper. We also changed the dGt/dt to the correct dM/dt and the respective units (if not unitless) are provided in table caption. We also changed the order of the columns to follow the order of Figures 6 and 8 for Experiment 1 and Experiment 2 respectively. Moreover, we now present a separate table for George VI and Larsen C basins and added the Coulomb sliding BISICLES simulation for Experiment 1 to Table S1.**

Page19

l4: You definitely need to show your results with the Fuss and Farinotti geometry

**The Huss and Farinotti results have been added to Figure 4**

Suplementary

Figure A3: Could you explain why you chose $\lambda_C = 10^{-1}$? It doesn't seem that obvious looking at A3c. Stupid question maybe, why is there a jump between $10^{-1}$ and $10^1$ (I mean no $10^0$ appearing)?

**Apologies. The exponent 0 was missing. This is now fixed. We chose $10^{-1}$ because we wanted to make sure that we are close to the kink of the L but still on the lower branch. $10^0$ was too close to the kink in our opinion.**

FigureA6: Time origin should be 2000

**Changed accordingly.**

**We amend the following sentence to the manuscript acknowledgements, to read: 'We thank the editor Olivier Gagliardini, Lionel Favier, and an anonymous reviewer for comments which improved the manuscript.'.**

Clemens Schannwell

---

## Referee Report (RR1)

**Review of the revision of "Dynamic response of Antarctic Peninsula Ice Sheet to collapse of Larsen C and George VI ice shelves" by Clemens Schannwell and his colleagues**

June 5, 2018

**General comments**

I now have re-read this revision of the Clemens Schannwell and colleagues's paper. Most of the comment that I did previously were followed by the authors and I am quite happy with this version.

---

## Author Response (AR2)

Schannwell et al. (2018, TCD) 'Dynamic response of Antarctic Peninsula Ice Sheet to potential collapse of Larsen C and George VI ice shelves.' MS No: tc-2018-21

**Response to final reviews (08-06-2018)**

We thank the reviewers for their additional comments which we have addressed as follows. Reviewer comments are in black regular font text. Our responses are in *black italic font text*, and revisions to the manuscript are shown in *blue italic font text*.

Report #1 (Anonymous Referee #1)

Recommend manuscript should be accepted subject to minor revisions.

1 Summary statement

The revised manuscript by C. Schannwell and co-authors addresses most of the points raised by both reviewers. The new version is clear and the authors tried to present a more detailed analysis of the results and an improved discussion. There is one thing that remains unclear and not really discussed: it concerns the similarity of the results between experiments 1 and 2 for the PSU3D model but not for the BISICLES model. The BISICLES results show a much larger grounding line retreat and ice loss in experiment 2 than experiment 1 for George VI ice shelf, while one would expect a larger response for experiment as the changes are more extreme (removal of all the ice shelf).

Note that the page and line numbers refer to the version of the manuscript including the highlighted changes.

2 Major comments

There is no real discussion on the differences between experiments 1 and 2. For Larsen, results are similar for the two experiments and for both PSU3D and BISICLES models. For George VI, the results are similar between experiments 1 and 2 for PSU3D but significantly different for BISICLES. The thinning rates and grounding retreat (Fig. 6 and 8) as well as the ice loss (Fig.4 and Fig.7) show a much larger response in for experiment 2 than for experiment 1, which is unlike one would expect as experiment 1 represents a much more radical change (removal of all the ice shelf). There is not much explanation about this, and the authors should at least discuss why the behavior of PSU3D is similar why the behavior of BISICLES is different for the two experiments, and also explain why the changes are larger for experiment 2 than experiment 1 in BISICLES.
*We think this is a misunderstanding. Experiment 1 does not necessarily represent a larger perturbation than Experiment 2. Owing to the crevasse calving law and their different implementations, Experiment 2 has much more potential to vary. In our simulations, either very little ice can calve (RCP4.5 scenario) or given enough surface water the entire shelf can collapse and new floating areas (that were formerly grounded) keep on calving. So unlike Experiment 1 where collapse is only enforced once, we can have repeated/continuing collapse of the shelf in Experiment 2. To make this clearer in the manuscript, we added:* *The discrepancy in sea-level rise projections between Experiment 1 and Experiment 2 is a result of the different applied perturbations. In Experiment 1, the entire ice shelf is removed at the start of the simulation before a fixed calving front is employed. In contrast, Experiment 2 with the crevasse calving law has much more potential to vary. Our simulations show that either very little ice can calve (RCP4.5 scenario) or given enough surface water the entire shelf can collapse and emerging new floating areas that*

*were formerly grounded keep on calving. So unlike Experiment 1 where collapse is only enforced once, repeated/continuing collapse of the shelf can occur in Experiment 2 (RCP8.5 BISICLES simulation).*

The reasons for the agreement between PSU3D and BISICLES for Larsen C and disagreement for George VI are outlined on page 16 in the paragraph starting at line 15 (highlighted MS) where it reads for Larsen C: *We attribute the good agreement across both models for Larsen C to the fact that the area of the marine-based sectors is limited in this domain (2.1 mm contained in marine-based sectors) due to the very mountainous bedrock topography constraining potential grounding-line retreat. This is supported by all simulations across all ice-sheet models as even under a wide range of different forcings the Larsen C embayment does not contribute more than 4.2 mm by 2300.* And for the differences in the George VI it reads: *As this large grounding-line retreat is only initiated in the BISICLES simulation, large differences in sea-level rise projections occur. The most likely explanation for this differing behaviour is due to the difference in the inferred basal traction coefficient fields that affects each model's response to ice-shelf removal. PSU3D predicts much higher-friction bedrock conditions in the George VI embayment than BISICLES (Figure 2). These high friction bedrock conditions result in little acceleration of the major outlet glaciers following ice-shelf breakup in Experiment 2. This in turn means that the calving law applied to only floating ice cells cannot drive the initial retreat into the marine based sectors as the outlet glaciers do not thin sufficiently to form floating ice tongues. In contrast in the RCP8.5 BISICLES simulation for George VI, speed-up in response to ice-shelf breakup leads to enhanced dynamic thinning of the main outlet glaciers. This thinning in conjunction with the calving law drives the calving front into the marine-based sectors where further retreat is initiated by a combination of the marine ice-sheet instability and the meltwater driven calving law, resulting in the simulated much higher sea-level rise projections.*

Another relatively minor point relates to the tables that were in the first version of this manuscript and disappeared into a supplementary material. As mentioned by both reviewers, the tables were not clear and needed some clarification, which does not mean to hide them. It would be best to put them in the Appendix with the supplemental figures, and make sure they are easy to understand.

*We have clarified the tables as suggested by both reviewers and had moved the table to the supplementary as suggested by the other reviewer because it disrupted the reading flow. We have now moved the table to the Appendix of the main manuscript and further clarified the table caption.*

3 Specific comments

p.1 l.4: "in future" → "in the future"

*Text changed to ..in the future.*

p.1 l.11: "relative importance to sea level of the large ..." → " relative importance of the large ... of collapse on sea level rise."

*We have rewritten this sentence according to the reviewer's suggestion but find that it no longer makes sense nor provides grammatically correct English. As such, we prefer to keep the sentence as before. The key proposition is the first part of the sentence, that we measure the 'varying and relative importance to sea level' of potential collapses.*

p.5 l.2: "A is the depth averaged rheological exponent" → "A is the depth averaged rheological

coefficient"

*Changed to read .. depth-averaged rheological coefficient.*

p.5 l.12: "the thickness" → "the ice thickness"

*Changed to read .. ice thickness.*

p.7 l.1: I still think this is confusing. You want the model to represent "reality", or the version we know of reality provided by observations. So the models "should" only be as close as possible to that. I understand the argument of being close to equilibrium to simplify the comparison between models and assess the impact of the ice shelf collapse, but the first sentence is really misleading and not accurate, and should be at least rephrased.

*We have changed this sentence to read: Following initialisation, the sheet-shelf models should aim to be as close to the steady-state initial conditions provided by observations, as long as the ice sheet itself is in steady state, such that dh/dt = 0.*

p.10 Tab.1: Add the default sliding on BISICLES Experiment 1 (third line of the table). Also add "zero melt" or something along this line for Experiment 2 (lines 8 and 9).

*Default sliding added to Table 1 line 3, zero melt added to lines 8 and 9.*

p.12 Figure 4 caption: How is the uncertainty in BAS-APISM computed?
We added: Uncertainties are quantified by a Monte-Carlo simulation (see Schannwell et al. 2016).

p.12 Figure 4 caption: "Upper panels show" and "Lower panels show"

*Text amended to read show, in both cases.*

p.16 l.23: "scenario" → "reason/explanation"

*Text amended to read explanation*

p.16 l.25: "sticky" → "high friction"

*Text changed to higher-friction and high friction here and nearby.*

p.18 l.14-20: This paragraph is not really convincing, the arguments should be more detailed.

*This paragraph is rewritten to now read: For Experiment 2, total maximum grounding-line retreat rates are similar to those from Experiment 1 when simulated by PSU3D. In the BISICLES simulation retreat rates increase from 6.4 km in Experiment 1 to 21.3 km in Experiment 2. This is in agreement with computed sea-level rise projections (Figure 7). Significant speed-up is absent in the years following ice-shelf removal across all basins due to the more gradual loss of buttressing in Experiment 2 (compared to the complete ice-shelf removal in Experiment 1). This results in a less dramatic dynamic response than in Experiment 1, with the exception of several basins of George VI Ice Shelf where retreat rates can lead to large mass losses. The gradual loss of buttressing simulated by Experiment 2 leads to grounding-line retreat and mass loss response occurring >15 years after ice-shelf removal.*

p.29 Figure A4 caption: "distribtution" → "distribution"

*Text changed to read distribution.*

p.31 Figure A6 caption: should be "Upper panels show" and "Lower panels show"

*Text amended to read show in both instances.*

Report #2 (Anonymous Referee #2)

Recommend manuscript should be accepted as is.

We look forward to hearing back positively.

Regards,

Clemens Schannwell and Nick Barrand, on behalf of the author team.

---

## Author Response (AR3)

Schannwell et al. (2018, TCD) 'Dynamic response of Antarctic Peninsula Ice Sheet to potential collapse of Larsen C and George VI ice shelves.' MS No: tc-2018-21

**Response to Editor review (22-06-2018)**

We thank the reviewers for their additional comments which we have addressed as follows. Reviewer comments are in black regular font text. Our responses are in *black italic font text*, and revisions to the manuscript are shown in *blue italic font text*.

- caption Fig 1: the red inset are not only used by Fig 8, but also Fig 2 (and others?)? Should be corrected.

We added all Figures in the main text that show the same zoom-in.

Also, what are the model domains should be specified (not here, but in the text)? Whole Antarctica? Same for all 3 models?

Added the following sentence about model domain extent: Model domains vary across the models with BAS-APISM and BISICLES including the entire Antarctica Peninsula and PSU3D simulating the Larsen C embayment and George VI embayment separately (red rectangles in Figure 1).

- page 5, line 2: by surface liquid, you mean water? Why not water then?

Changed

- page 5, line 25: "is to (Pollard et al., 2015)." -> "is to (Pollard et al., 2015):".

Changed

Also, R in Eq. 11 is not defined (at least its definition is not clear from line 25.

We edited the mentioned sentence to make it clearer. It reads: This formula scales surface melt (R in equation 11) exponentially with mean DJF near-surface temperatures and approximates the surface melt available to fill surface crevasses.

- caption Fig. 3: Give in words what is dV/dt

Changed to "Volume change ..."

- caption Fig. 4: "Note different" -> "Note the different" (?)

Changed accordingly

- Figure 7: In the y-axis label, missing a space after loss and (%); and loss and (km2). Also, there should be 4 curves in panels e and f as in the 4 other panels? It seems that there is only 3?

Thanks for spotting this. The lines were present in the initial submission but changes to the plotting script made them disappear. Changed accordingly.

- page 15, line 8: "Tables A3,A4" -> "Tables A3 and A4"

Changed here and throughout.

- page 16, line 21: "this boundary condition results..." it is not clear what did you change? Is it the bed DEM? Then it should be specificaly mentioned instead of using "boundary condition" which can also refer to sliding, etc... Same line 24 for boundary input dataset. Is it the bed DEM dataset that you are refereeing to?

**You are correct. We changed this to "bedrock topography" in both instances**

- page 17, line 5: an analysis using the five drainage basins is not really done in section 3.4, as given number stay general. To justify the introduction of these 5 basins at the stage of the paper, I would expect that the discussion in this section is really focussed on the differences observed from one to an other basin. On the same line, dh/dt presented in Fig. 8 are not really discussed, especially the differences between one basin to an other. Only the tables in the Annex give explicit number for each individual basin. Fig 8 and Tables A1 to A4 should be commented a bit more in this section?

This section was introduced to compare our projections to the past ice-shelf collapse of Larsen B Ice Shelf and the subsequent retreat pattern of the tributary glaciers. However, we agree that this section needed a bit more substance. So we expanded the section to accommodate more results from the Tables and Figure 8. The section now reads: To further assess the impact of ice-shelf break-up, five drainage basins from the Larsen C embayment (LarI-LarV, Figure 8) and George VI embayment (Geol-GeoV, Figure 8) were selected for additional analysis. This provides a comparison to real- world examples of the magnitudes and pattern of glacier response to ice-shelf collapse. For Experiment 1 (immediate ice-shelf collapse), the speed up following ice-shelf removal is short lived (~15 yrs) for both models. Maximum speed-up of ~300% is possible, though the mean maximum speed-up is ~50% (Tables A1 to A4). These values are smaller than those observed following Larsen B collapse with a maximum of 8 fold speed up (Rignot et al., 2004). This may be due to the different areas selected for the speed up calculation. Both rates of ice discharge (mass loss) and grounding-line retreat are greatest immediately following shelf collapse. For 65% of the selected 10 drainage basins, more than 50% of the total modelled grounding-line retreat takes place within 15 years of ice-shelf collapse. Maximum mass loss rates for Larsen C (1.6-5.1 Gt a-1) are smaller than observations for a similar time period for Larsen B (8.0 Gt a-1) (Scambos et al., 2014). Responses of individual drainage basins for Larsen C and George VI are highly variable with grounding-line retreat ranging from 3.7 km to 26.8 km for Larsen C and 3.1 km to 12.2 km for George VI (Tables A1 and A2). This high spatial variability in response across the selected basins indicates that the importance of ice-shelf buttressing is also highly spatially variable. Most of the grounding-line retreat, in particular for Larsen C,

occurs in areas of bedrock channels (Figure 5). Since these deep bedrock channels are smaller for Larsen C, this leads to smaller mass loss than for George VI even though maximum grounding-line retreat numbers are larger. But grounding-line retreat is spread across a wider area of the drainage basin front at George VI (Figure 6).

For Experiment 2, maximum grounding-line retreat (4 km to 33.4 km, Tables A3 and A4) is of similar magnitude to Experiment 1 (3.1 km to 26.8 km, Tables A1 and A2) including the spatial variability across the selected basins for PSU3D in both embayments. However, for BISICLES this holds only true for the Larsen C embayment where mass loss averaged across the five basins remains at 0.1 Gt a-1 for both experiments. For the George VI domain, the spatial variability across the basins is strongly reduced with maximum grounding line retreat now ranging from 10 km to 25.5 km (Table A4). When averaged over the five basins grounding-line retreat increases from 6.4 km for Experiment 1 to 21.3 km in Experiment 2 for George VI. The retreat in this experiment spreads over the entire width of the drainage basin front (Figure 8d), resulting in an increase of mass loss over the 300 years from 0.5 Gt a-1 in Experiment 1 to 3.0 Gt a-1 in Experiment 2. This is in agreement with computed sea-level rise projections (Figure 7). Significant speed-up is absent in the years following ice-shelf removal across all basins due to the more gradual loss of buttressing in Experiment 2 (compared to the complete ice-shelf removal in Experiment 1). This results in a less dramatic dynamic response than in Experiment 1, with the exception of several basins of George VI Ice Shelf where retreat rates can lead to large mass losses. The gradual loss of buttressing simulated by Experiment 2 leads to grounding-line retreat and mass loss response occurring >15 years after ice-shelf removal.

- page 17, line 9: "8fold speed up" -> "8 fold speed up"

**Changed**

- page 17, line 12: each of this mass loss rates number should be put in context of the maximal potential sea level contribution of each sector? The same apply at other place of the paper, especially in the predicted contribution of Larsen C relative to George VI?

We do not understand how we should relate this maximum mass loss rate for the selected drainage basin to the maximum potential sea-level contribution. The aim of this comparison with observations from Scambos et al. 2014 for Larsen B was to see if mass loss rates are in a similar range, which they are. However, as raised below we have now added in the conclusion that of the marine-based ice in Larsen C is lost in our experiments which in comparison is much larger than for George VI where this is <=55%. This now reads: This is in contrast to the Larsen C embayment where the majority of the projections exceed 55% with the highest projections exceeding 100% of the grounded ice that is located below sea level. This in turn means that sea-level rise induced by ice-shelf removal is limited by the small area of marine-based sectors in the Larsen C embayment.

- page 17, line 15: what does 6.4 km represents? retreat rate of what, the grounding line? It should be mentioned. Also, with the units of km, it is not a rate of retreat. I guess it should be : "the simulated retreat of the GL over XX years is 6.4 km in ..."?

Yes, we have corrected this here and throughout the manuscript. It should indeed be grounding-line retreat and not grounding-line retreat rate.

- page 19, line 3: give at which date these sea level projections apply.

**We added: by 2300**

- page 19, line 5: I don't understand the "only" 55% as it seems already a large number if you have loss 55% of the total potential contribution to sea level rise? As in my previous remarks, this is an interesting number that should be given also for Larsen C?

As mentioned above we are giving the same number for Larsen C now. We would like to keep the "only" because we want to underline the fact that there is another 50% of ice in these drainage basins that may be vulnerable to rapid ice-sheet retreat.

- Figures A1 and A2 could be smaller (one column)

Done

- caption Table A1: for dGL\_15, it is not really averaged over 15 years but it gives the total retreat over 15 years?

Yes, thanks for spotting this. We adjusted the table caption to account for this.

- captions of Tables A2 to A4 could be simplified : "Same as Table A1 but for George VI embayment.

Done

**Clemens Schannwell**

[revised manuscript text omitted]